# Bacimethrin, an allelopathic vitamin B$_1$ antagonist, is linked with microbial gene expression patterns in a hypereutrophic watershed

Kelly C. Shannon[1], Frederick S. Colwell[1,2], Byron C. Crump[1,2], Elizabeth Brennan[1], Gillian St. John[1], Robin Gould[3], Christopher Hartzell[3], McKenzie Wasley[4], Christie Nichols[4], Clifford E. Kraft[5], Christopher P. Suffridge[1]*

1 Department of Microbiology, Oregon State University, Corvallis, Oregon, 2 College of Earth Ocean and Atmospheric Sciences, Oregon State University, Corvallis, Oregon, 3 Department of Biochemistry and Biophysics, Oregon State University, Corvallis, Oregon, 4 U.S. Fish and Wildlife Service, Klamath Falls Office, Klamath Falls, Oregon, 5 Department of Natural Resources, Cornell University, Ithaca, New York, United States of America

* suffridc@oregonstate.edu

## Abstract

Freshwater cyanobacterial harmful algal blooms (cyanoHABs), often dominated by *Aphanizomenon*, *Dolichospermum*, and *Microcystis,* are intensifying in eutrophic watersheds globally. A potential control on bacterioplankton dynamics in these systems is the availability of the essential metabolic cofactor thiamin (vitamin B$_1$) and presence of the allelopathic thiamin antagonist bacimethrin, which causes competitive inhibition of thiamin-requiring enzymes. We examined dissolved concentrations of thiamin chemical congeners and bacimethrin, 16S amplicon-based microbiome compositions, prokaryotic mRNA-based metatranscriptomes, and reference genomes in hypereutrophic Upper Klamath Basin before and during seasonal cyanoHABs. Our objective was to connect bacterioplankton community compositions and gene expression patterns with thiamin congener and bacimethrin availability under different cyanoHAB conditions. Bacimethrin was present in all samples at similar concentrations to the thiamin precursor, HMP, suggesting that similar mechanisms influence the availability of both compounds. Additionally, bacimethrin concentrations were positively correlated with cyanoHAB species abundance (cells mL$^{-1}$) and the expression of microbial thiamin biosynthesis genes. Samples with high cyanoHAB abundance also displayed elevated transcription of genes in key biochemical pathways such as the pentose phosphate pathway, photosynthesis, and glycogen biosynthesis. Bacterioplankton such as *Limnohabitans* spp. that are unable to synthesize thiamin, and are thus vulnerable to bacimethrin allelopathy, showed reduced gene expression when cyanoHAB abundance was high. Reference genomes of cyanoHAB and many picocyanobacteria strains contained complete thiamin biosynthesis gene pathways, implicating these taxa as major thiamin

**Data availability statement:** All microbial sequence processing, raw LCMS data, and data analysis scripts can be found here: https://github.com/ksmicrobe/UKB-Thiamin/tree/main/Manuscript_Code and all sequencing data (MiSeq and NextSeq) can be accessed with the following NCBI SRA BioProject ID: PRJNA1216032.

**Funding:** This work was funded by United States Fish and Wildlife Service grant F22AC01810-01 and California Department of Fish and Wildlife grant Q2196012, both to Christopher P. Suffridge. Additional personnel funding for Christopher P. Suffridge was provided by National Science Foundation grant DEB-1639033. Mass spectrometry instrumentation at the OSU Mass Spectrometry Center was supported by National Institutes of Health grant 1S10RR022589-1.

**Competing interests:** The authors have declared that no competing interests exist.

sources. These results suggest that bacimethrin provides a competitive advantage to bacterioplankton that do not require exogenous thiamin by eliminating the risk of bacimethrin uptake with thiamin transporters, potentially facilitating cyanoHAB dominance in Upper Klamath Basin and broader eutrophic watersheds.

## Introduction

Discovering mechanistic drivers of microbial interactions is a central goal in microbial ecology, particularly in aquatic ecosystems where diverse bacterioplankton communities mediate regional to global scale biogeochemistry, shaping ecosystem function at large scales [1]. Recent advances in meta-omics have greatly accelerated this effort by integrating environmental data, including dissolved metabolite profiles, into large scale correlative analyses of microbial community composition and function [2–4]. This approach enables more direct inferences of ecological processes and facilitates the generation of hypotheses about the underlying mechanisms that drive correlative trends. In this study, we sought to describe bacterioplankton interactions and biogeochemistry in hypereutrophic freshwater environments by combining dissolved measurements of the essential metabolite thiamin, its chemically related congeners, and a newly-measured toxic thiamin antagonist with microbial community data.

The complex biosynthesis and exchange of thiamin, its chemical congeners, and bacimethrin (thiamin and related compounds; TRCs hereafter) provide an ideal framework for examining microbial interactions. Thiamin is required across all domains of life and microbial communities produce the bulk of thiamin for aquatic food webs [5,6]. Paradoxically, given its broad necessity, most microbes salvage TRCs from the dissolved pool to supplement thiamin biosynthesis [7–9]. As such, it has been shown that the dissolved availability of these essential TRCs can structure bacterioplankton communities [10,11]. Thiamin is a coenzyme that, in its active form of thiamin pyrophosphate (TPP), is required by enzymes key to catabolic and anabolic pathways of carbon metabolism including the tricarboxylic acid cycle, the Calvin cycle, branched chain amino acid biosynthesis, and the pentose phosphate pathway [9].

Many bacterioplankton species are thiamin auxotrophs, lacking the full *de novo* genomic pathway for producing thiamin [7]. Auxotrophic bacterioplankton must import exogenous thiamin or thiamin precursors including the pyrimidine 4-amino-5 hydroxymethyl-2-methylpyrimidine (HMP) and the thiazole 5-(2-hydroxyethyl)-4-methyl-1,3-thiazole-2-carboxylic acid (cHET) [12] to meet cellular thiamin requirements [7], with pyrimidine (HMP) auxotrophies being the most prevalent [7,13–15]. Thiamin is also abiotically degraded into 4-amino-5-aminomethyl-2-methylpyrimidine (AmMP; pyrimidine degradation product) and 4-methyl-5-thiazoleethanol (HET; thiazole degradation product), which some organisms can recycle to synthesize thiamin [8,13,16,17]. In aquatic ecosystems, the net exchange of TRCs between producers (typically prototrophs that have the full *de novo* thiamin biosynthesis pathway) and

consumers (typically auxotrophs) controls dissolved TRC concentrations [13,18], which, when coupled with microbial community data, provides a framework to understand microbial thiamin cycling [19].

Bacimethrin, an understudied TRC with allelopathic properties, has been identified as a microbial secondary metabolite produced by cultured soil bacteria [20–22] and, likely, microbes in other environments. As a toxic analog of HMP [20,21], bacimethrin targets thiamin auxotrophs that lack HMP biosynthesis genes by interrupting proper thiamin biosynthesis. The incorporation of bacimethrin rather than HMP into the pyrimidine branch of the thiamin biosynthesis pathway leads to the formation of methoxy-thiamin pyrophosphate [23,24], a dysfunctional coenzyme that competitively inhibits thiamin-dependent enzymes, thereby disrupting essential metabolic functions. Although extensively characterized in laboratory cultures, the ecological role of bacimethrin in natural environments remains poorly understood. In aquatic ecosystems, bacimethrin may alter bacterioplankton community structure by mimicking HMP and interfering with its uptake via HMP transporters [25] and use via thiamin biosynthesis enzymes [26]. Recent evidence shows that *Microcystis* spp., which can form cyanobacterial harmful algal blooms (cyanoHABs), can synthesize bacimethrin [27]. As thiamin prototrophs [9,28,29], *Microcystis* spp. are likely resistant to bacimethrin allelopathy. CyanoHAB-produced bacimethrin may also confer a selective advantage to prototrophs by competitively inhibiting thiamin or HMP-utilizing enzymes in auxotrophic taxa. This mechanism could facilitate cyanoHAB niche expansion and persistence through the disruption of native HMP auxotrophic microbial assemblages.

The potential niche expansion by cyanoHABs – via the competitive exclusion of bacimethrin-sensitive bacterioplankton – is of substantial global importance because the frequency and severity of cyanobacterial harmful algal blooms in freshwater ecosystems has increased [30] alongside anthropogenic stressors, such as increased water temperature, nutrient pollution, agricultural and industrial activity, and urbanization [31,32]. CyanoHABs magnify eutrophication by producing toxins (cyanotoxins) and altering aquatic food web stability [33,34], with impacts [35] that cost billions of dollars annually to manage due to their detrimental impacts on freshwater ecosystems [30,36]. CyanoHABs have been extensively studied in Upper Klamath Basin (UKB) [37,38]. Sediment cores from Upper Klamath Lake (UKL; the lake only rather than the whole basin) indicate that increases in cyanoHAB species abundance in the early 20th century corresponded with an intensification of eutrophication, which was driven by increased N and phosphorus (P) inputs into the lake from previously drained marshlands and agricultural activity [39]. Over decadal times scales, cyanoHAB species have also gradually replaced diatoms in UKL during summer months [40]. Contemporary UKL cyanoHABs, which can also impact other UKB habitats (rivers and reservoirs), are typically initiated by blooms of filamentous and diazotrophic (can fix atmospheric nitrogen; $N_2$) *Aphanizomenon* and *Dolichospermum* spp. [41]. These cyanoHAB species then grow rapidly during the early-summer and reach peak biomass in June or July, corresponding to an annual peak and subsequent rapid decline in water pH, dissolved oxygen (DO), and primary production [41]. Blooms of *Microcystis* generally follow filamentous cyanoHAB species and have similarly been linked to habitat degradation in UKL [42,43].

We investigated connections between the dissolved availability of TRCs (thiamin, HET, cHET, AmMP, HMP, and bacimethrin) and bacterioplankton communities (including cyanoHAB species) in UKB by pairing measurements of TRC concentrations with 16S rRNA gene-based analysis of microbiomes, prokaryotic mRNA-based metatranscriptomes, and reference genome assemblies. Samples were collected prior to (May) and during (August) a seasonal cyanobacteria bloom in UKL and its springs, tributaries, outflow, and surrounding reservoirs. We hypothesized that the abundance of cyanoHAB species would influence dissolved TRC concentrations and patterns of microbial gene expression. Therefore, we sought to (1) investigate the environmental distributions of TRCs across diverse UKB habitats, (2) identify UKB habitats, cyanoHAB-related seasons (prior to and during cyanoHABs formation: spring and summertime), and conditions associated with the greatest potential for bacimethrin allelopathy to bacterioplankton, and (3) link patterns of bacterioplankton gene expression and thiamin biosynthesis gene activity with TRC concentrations in water samples with contrasting impacts (high & low) of cyanoHABs.

## Materials and methods

### Sampling procedure, sample site description, and water quality measurements

Microbial and TRC surface water samples were collected May 1–3 and August 28–30, 2023, during daylight hours following previously described methods [18]. Sampling sites were selected to examine variable impacts of cyanoHABs and to contrast lotic (UKL tributaries and outflow) and lentic (UKL and reservoirs) systems (Fig 1A; see S1 Methods for sample site description). In UKL, *Aphanizomenon* and *Dolichospermum* spp. blooms typically begin in late May and *Microcystis* spp. blooms begin in late July, after which both blooms can persist into late Fall [44]. Thus, samples were binned into pre-bloom (early-May) and bloom (late-August) periods. Publicly available USGS data (https://waterdata.usgs.gov/nwis) for UKL water elevation, Sprague and Williamson River discharge, and Link River pH and DO were accessed and plotted with the dataRetrieval USGS R package [45] (see SI).

Samples were collected at ~0.5 m depth using 1 L amber HDPE bottles (Nalgene). All samples were prefiltered to remove metazoans and large particles (100 μm mesh filter) and stored on ice until further processing. Within the same day of sampling, peristaltic filtration at a rate of approximately 30 mL min$^{-1}$ was used to collect cells and particles onto

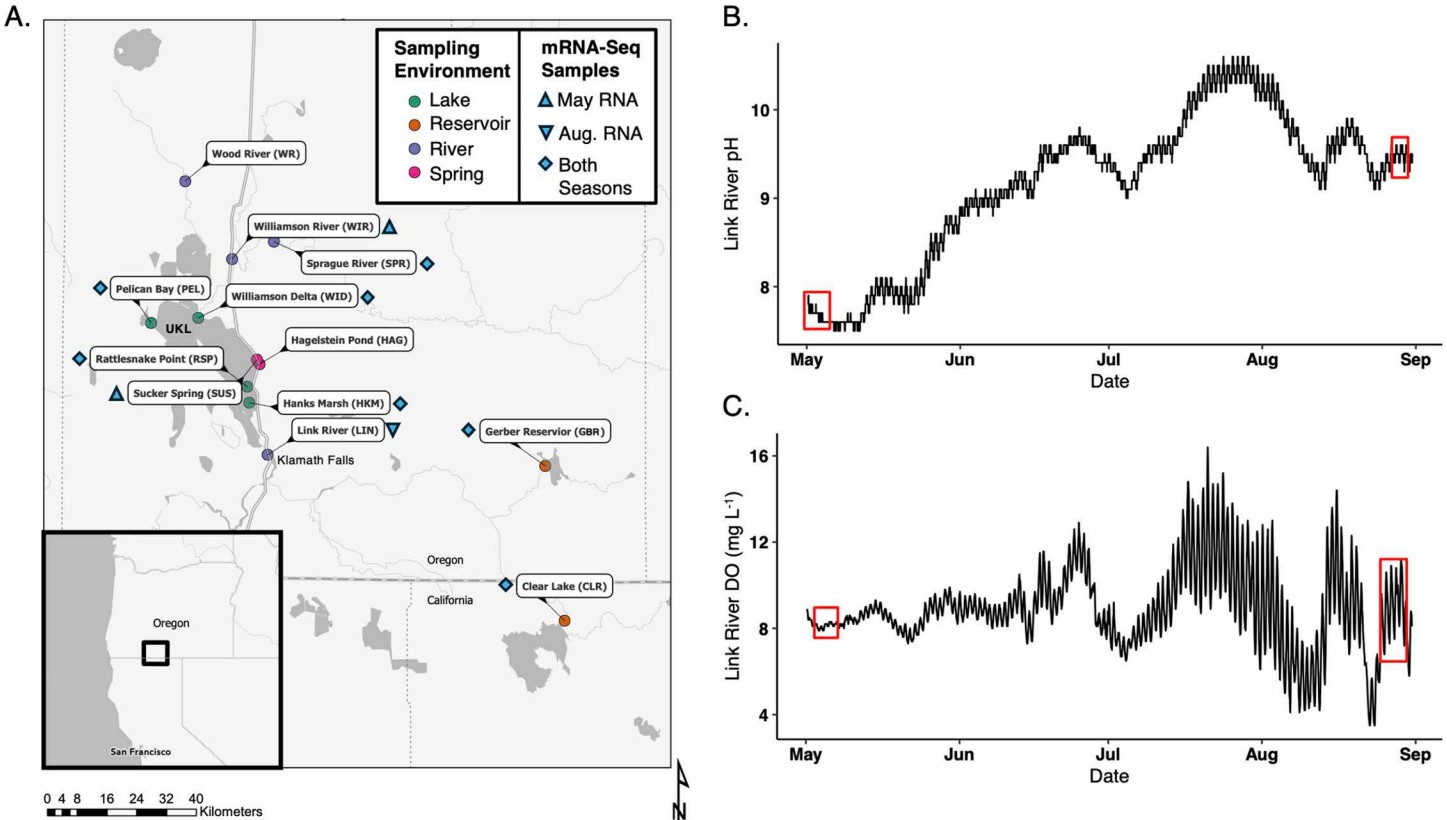

**Fig 1. A map of sampling locations and ecological conditions in Upper Klamath Lake between late spring and summer. (A)** Sample map showing the locations of sample collection. The environment type is indicated by color of sampling points. Light blue shapes indicate if sufficient RNA was extracted from samples to produce metatranscriptomes and, if so, which seasons are represented by metatranscriptomes at each sample site; May=pre-bloom and Aug.=bloom time periods. Sample abbreviations shown here apply to subsequent figures. Analyses for 16S, dissolved thiamin congener and dissolved bacimethrin were conducted at all locations and time points. USGS continuous water quality measurements of (B) pH, (C) dissolved oxygen measurements at a Link River USGS water quality station (SI) were compiled and plotted between May 5th and Sept. 1st, 2023. Red boxes indicate the range of sampling dates for DNA, RNA, and TRC measurements.

0.22-µm Sterivex filters (PES membrane, Millipore, Burlington, MA, U.S.A.) and volumes of filtered water were recorded. Immediately following filtration, 1 mL of RNA*later* (Thermo Fisher Scientific) was pipetted directly into Sterivex cartridges, which were then sealed and incubated in the dark for 5 minutes. Following incubation, sealed Sterivex cartridges were flash frozen in liquid nitrogen and immediately placed in a −80°C freezer until analysis.

Cell-free filtrate was used for TRC and nutrient concentration measurements. TRC samples were collected in acid-washed and methanol-rinsed amber HDPE bottles, acidified with 1 mL of 1 M hydrochloric acid, and stored at −20°C until analysis at Oregon State University (OSU). Nutrient samples were collected in acid washed glass vials, frozen at −20°C and shipped to the University of California, Santa Barabara Marine Science Institute Analytical Lab for analysis using their flow injection system for nitrate + nitrite, ammonium, and phosphate. Temperature (°C) and DO (mg L$^{-1}$) were measured with a Thermo Fisher Scientific (Waltham, MA, U.S.A.) Orion multiparameter meter with RDO and temperature sensors. Water quality conditions were predicted using the Kann and Walker [46] surface water elevation model. This model predicts the probability of poor water quality conditions based on surface elevation of the lake, where deviations from normal surface elevations (based on historical data) will predict high likelihoods of poor water quality conditions.

### Dissolved TRC analysis

Dissolved TRCs were extracted from water samples and analyzed using LC-MS as previously described [18,27]. Briefly, samples were thawed, their pH was adjusted to 6.5, and TRCs were extracted using $C_{18}$ resin (Agilent Bondesil HF). TRCs were eluted from the $C_{18}$ resin using 12 mL of methanol, then further concentrated by nitrogen drying to a volume of 250 µL. A 1:1 chloroform liquid phase extraction was used to remove hydrophobic compounds from the sample matrix. Analysis was conducted using an Applied Biosystems 4000 Q-Trap triple quadrupole mass spectrometer with an ESI interface coupled to a Shimadzu LC-20AD liquid chromatograph (see S1 Methods). Chromatography and mass spectrometer conditions for all TRCs are described elsewhere [18,19,27]. Bacimethrin was observed to have column retention time of 2.89 minutes, a parent m/z of 156.1, and daughter product m/z of 138.0, 95.0, and 81.0. The declustering potential used was 20 V and the collision energy was 25 V. The limit of detection for bacimethrin was observed to be 2.21 nM, which was calculated as three times the standard deviation of the lowest standard used in analysis [27]. Environmental values are reported in the pM range while the LOD is in the nM range due to the approximately 4 order of magnitude concentration factor produced by the SPE procedure [19]. Samples were analyzed in triplicate and were randomized prior to analysis. An internal standard ([13]C-labeled thiamin) and external standard curves were used for calibration and quantification. The inclusion of the internal standard allowed for concentrations to be corrected for matrix effects. The TRC values presented in this manuscript are the means of three technical replicates. The standard deviations and means of these replicates are presented in S1 Table. Analysis was conducted at the OSU Mass Spectrometry Center. A subset of the bacimethrin concentration data (Hanks Marsh and Williamson River sites) (Fig 1A) have been previously published by [27].

### Microbial DNA and RNA extractions and sequencing

All DNA and RNA extraction and purifications were performed with ZymoBIOMICS (Irvine, California, U.S.A.) DNA/RNA Miniprep Kits in a biosafety hood that was sequentially sterilized with UV radiation, Obliterase (Innovative Scientific Solutions; Dayton, Ohio, U.S.A.), and 70% ethanol (see S1 Methods). Manufacturer instructions were followed except for minor changes to the initial bead-beating procedure (see SI). RNA samples were further purified and concentrated with ZymoBIOMICS clean and concentrator kits following manufacturer instructions except that the in-column DNase I treatment was performed only once, following the first cleaning step, and RNA was eluted with 30 µl DNase/RNase-free water. Internal standards were added prior to extraction for DNA absolute abundance measurements (SI for full details) [47].

For DNA, PCR was performed with Platinum II Hot-Start polymerase (2x) (Invitrogen; Waltham, Massachusetts, U.S.A.) to amplify the prokaryotic V4 region of the 16S rRNA gene with 2.5 µl of 2 µM 515F-806R primers [48]. 16S amplicons

were sequenced with Illumina (San Diego, California, U.S.A.) MiSeq 2x250 bp paired-end high throughput sequencing (HTS) (SI for PCR protocol and MiSeq preparation). Purified and concentrated RNA was sent to OSU Center for Quantitative Life Science (CQLS) for Illumina NextSeq 2x100 bp paired-end HTS, following sequencing preparation with the Illumina Ribo-Zero Plus rRNA Depletion Kit by the CQLS. Following Illumina preparation by the CQLS, 16 of the 24 samples had enough RNA for sequencing (Fig 1A). While this did technically bias results towards samples with higher RNA yields, these 16 samples still represented all ecosystem types aside from spring water sites, which likely contained microbial biomasses that were too low to yield sufficient mRNA to be detected following the full mRNA workflow: extraction, cleaning and concentrating, Illumina preparation, and NextSeq.

## DNA sequencing data bioinformatics and statistics

Statistical analyses were performed in RStudio (v4.2.1) and code for the generation of all DNA and RNA figures can be found on GitHub (Data Availability Statement). 16S rRNA gene sequencing data was demultiplexed by the CQLS and processed following previous methods [18]. Briefly, adaptor trimming and initial quality filtering (sequences with Phred scores <20 were dropped) was performed with TrimGalore (v0.6.6) [49] and read processing to amplicon sequence variants (ASVs) was performed with DADA2 (v1.24.0) [50] following default parameters aside from the inclusion of pseudo pooling during ASV assignment.

To improve comparability between 16S and metatranscriptome taxonomic results, the Genome Taxonomy Database (GTDB; v202) [51] was used for 16S taxonomic annotations other than those for *T. thermophilus*-based absolute abundance calculations (SI for detailed *T. thermophilus* steps). All 16S community composition and beta-diversity plots were made with the phyloseq (v1.42.0) and microViz (v0.11.0) packages [52,53]. Aitchison distance (Euclidean distance between samples after a center-log ratio (CLR) transformation) was used for compositional and beta-diversity analyses [54] and principal coordinate analysis (PCoA) was used to visualize community differences in ordination space. We used 16S rRNA gene copy numbers, which can vary in number between the chromosomes of different bacterial and archaeal species and higher taxonomies [55], of *Aphanizomenon*, *Dolichospermum*, and *Microcystis* spp. from the University of Michigan ribosomal RNA database (rrnDB) [56] to infer putative cyanoHAB species cellular abundances (cells mL$^{-1}$; more information in SI) [57]. These data were used to bin samples into high and low cyanoHAB abundance (high: > 1,000 cells mL$^{-1}$ *and* during bloom time period; low: all other samples). Though this threshold was somewhat arbitrary, the 75th percentile value of cyanoHAB abundance was 1,453 cells/mL and all six samples that fell under the category of high cyanoHAB abundance were above this 75th percentile value.

Microbiome and metatranscriptome compositional differences between statistical groups were performed with PERMANOVA, multivariate homogeneity of group dispersions, and ANOSIM tests (significance based on uncorrected *p*-values) with the vegan package (v2.6-4) [12]. Betadisper and permutest functions in vegan yielded a significant *p*-value ($p < 0.05$) for the two groups binned by cyanoHAB abundance, indicating significantly different dispersion of microbial communities between the groups. Therefore, as has been previously recommended [58], ANOSIM was used to test for community compositional differences between groups of high and low cyanoHAB abundance in taxa of microbiomes and metatranscriptomes. PERMANOVA was used to test for significant microbiome and metatranscriptome differences between samples taken in each season (pre-bloom in May and during the bloom in August) and between the four environmental types (spring water, river, lake, and reservoir).

## mRNA-seq data bioinformatics and statistics

Demultiplexed metatranscriptome reads were uploaded to the Cmbio (Germantown, Maryland, U.S.A.) microbiome bioinformatics HUB ("Cosmos-Hub"; see SI for a more detailed description of Cosmos-Hub taxonomic and functional annotation pipeline) for taxonomic and functional profiling and linear discriminant analysis (LDA) effect size (LEfSe) analysis [59]. A full description of functional and taxonomic mapping and metatranscriptomics statistical methods (including LEfSe)

performed with Cosmos-Hub can be found at: https://docs.cosmosid.com/docs/about (or under the "Documentation" tab from the Cosmos-Hub main website). Transcript functions were annotated with Enzyme Commission, Gene Ontology (GO) terms, Protein Family (Pfam) hidden Markov model-based, and MetaCyc metabolic reconstruction databases, which are all integrated in the Cosmos-Hub. LEfSe analyses were performed in the Cosmos-Hub to identify transcripts, grouped by taxa (from taxonomic pipeline) and function, that were significantly related to cyanoHAB abundance (p-values were uncorrected). Reference genomes (those that mRNA mapped to; basis of taxonomic annotations) of strains that were positively correlated with cyanoHAB abundance were accessed in Cosmos-Hub and searched for thiamin-related genes (SI).

Per-sample functions of interest (GO terms and thiamin-related Pfam annotations) were exported as normalized counts (copies per million; CPM) and manually curated to assign higher-level functional descriptions (SI). All read normalization was performed within the Cosmos-Hub pipeline (see Cosmos-Hub documentation). To examine the up- and down-regulation of gene expression in high and low cyanoHAB abundance bins, functionally annotated transcript CPM were corrected to z-scores [60], which were averaged across samples in each cyanoHAB abundance bin. p-values were adjusted to false discovery rate q-values indicating differential gene expression ($q < 0.05$) from per-sample z-scores with the pnorm function in R (two-sided tests; Data Availability statement for code). Metatranscriptome-based community composition and beta-diversity plots were produced following the same methods as for 16S taxonomy with microViz. For statistical tests including redundancy analysis and Spearman correlations between gene expression and TRCs and cyanoHAB abundances, the functionally annotated CPM abundance matrix was CLR-transformed (see SI for full analyses steps). PERMANOVA, ANOSIM, betadisper, and permutest functions were used following the same steps as for 16S taxonomy to assess between-group differences in transcriptionally active bacterioplankton.

## Results

### Dissolved TRC concentrations, nutrients, and bacterioplankton abundance

Median concentrations of dissolved TRCs (thiamin, HET, cHET, AmMP, HMP, and bacimethrin; i.e., not particle or cell associated) varied by 1–2 orders of magnitude across all UKB sample sites, and concentrations of thiamin and thiazole congeners were 1–3 orders of magnitude greater than those of pyrimidine congeners, including bacimethrin (Fig 2A; S1 Table; see Supp. Material for extended description of supp. tables). Mean and median picomolar (pM) concentrations of most TRCs in May (pre-bloom time period, or pre-bloom period) differed from those in August (bloom time period, or bloom period) and greatly varied by sampling environments, being lowest in spring water and highest in reservoir water for each TRC (Fig 2; Tables 1 and S1). Based on Wilcoxon signed-rank tests, mean concentrations did not significantly differ between the pre-bloom and bloom periods of any TRCs ($p > 0.05$). Mean TRC concentrations of thiamin, pyrimidine congeners, and bacimethrin were highest during the bloom period, while mean concentrations of thiazole congeners were highest during the pre-bloom period. The high TRC interquartile ranges in rivers relative to other UKB environments was driven by the low TRC concentrations observed in the Wood and Williamson Rivers compared to the Sprague and Link Rivers (Fig 2A; S1 Table). The Link River (UKL outflow) also showed much higher concentrations of pyrimidine TRCs, including bacimethrin and especially during the bloom period (Fig 2A; S1 Table).

Unlike TRCs, ratios of dissolved inorganic nitrogen (nitrate + nitrite + ammonium) to dissolved inorganic phosphorus (phosphate; N:P) differed greatly between pre-bloom and bloom periods (S1 Fig). During the pre-bloom period, 8 out of 12 samples showed N:P values less than the canonical Redfield ratio of 16:1, suggesting N limitation (S1 Fig) [61,62]. During the bloom period, all samples except Clear Lake displayed N:P far above the Redfield ratio, suggesting P limitation on primary production, though typical N:P ratios in lacustrine environments are higher than those in the ocean [63,64]. During the bloom period, ammonium concentrations were high and phosphate concentrations were low (S1 Fig).

Bacterioplankton absolute abundances (total 16S gene copies mL$^{-1}$ of sampled water) did not correlate with N:P concentration ratios (Spearman p-values > 0.05). Instead, bacterioplankton absolute abundances and TRCs each followed similar trends in magnitude across sampling environments, with increasing TRC concentrations and bacterioplankton

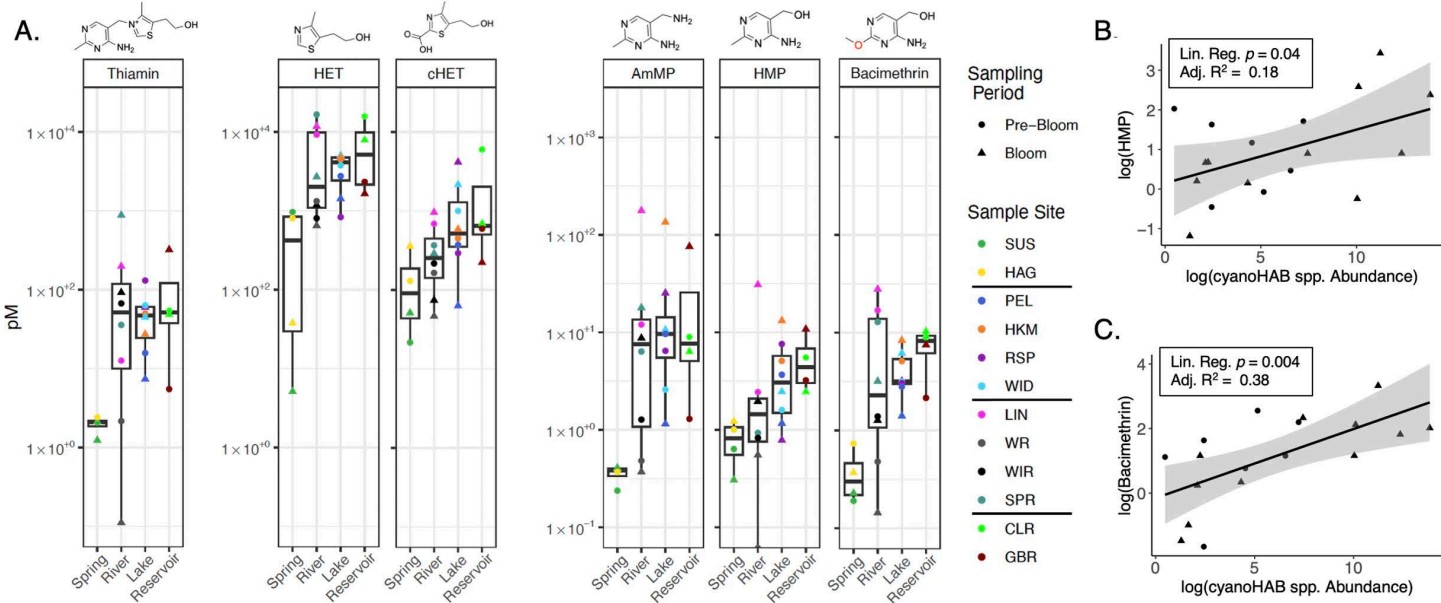

**Fig 2. TRCs vary across sampling environments and correlate with the abundance of cyanoHAB spp. (A)** TRC concentrations (picomolar; pM) presented as both discrete measurements and summary boxplots binned by sample environments. Lines are placed in between sample sites in color key to distinguish sampling environments (same order as shown in panel A x-axis). Chemical structures of TRCs are displayed above each compound's boxplot. Scales differ between thiamin, thiazole, and pyrimidine TRCs to allow for comparability between individual sample points and median concentrations under the pyrimidine TRCs. Scatterplots displaying linear associations between log-transformed cyanoHAB spp. abundance (cells ml⁻¹) and log-transformed **(B)** HMP and **(C)** bacimethrin concentrations. Linear regression results (*lm* in R function) are shown in boxes. In all panels, shapes correspond to sampling time periods: "Pre-Bloom" = May 2023; "Bloom" = August 2023.

**Table 1. TRC concentration (pM) differences between the pre-bloom and bloom periods.**

| TRC | Pre-bloom med. | Pre-bloom mean | Bloom med. | Bloom mean |
|---|---|---|---|---|
| Thiamin | 25.7 | **36.7** ± 38.7 | 46.6 | **140.2** ± 251.9 |
| HET | 2538.0 | **4857.4** ± 5839.3 | 2169.5 | **3478.5** ± 3545.8 |
| cHET | 369.4 | **858.3** ± 1643 | 322.9 | **802.3** ± 1207.4 |
| AmMP | 4.5 | **5.0** ± 4.4 | 9.7 | **38.4** ± 59.7 |
| HMP | 2.0 | **2.7** ± 2.4 | 2.0 | **5.6** ± 9 |
| Bacimethrin | 2.9 | **4.8** ± 5.4 | 3.2 | **5.8** ± 7.8 |

*Sample standard deviations are displayed after bolded mean values; "med." = median.*

absolute abundances from spring water to reservoirs (median lines in Fig 2 boxplots and S2 Fig). Based on Pearson correlations, all bacterioplankton absolute abundances were also significantly and positively correlated with concentrations of all TRCs ($p < 0.05$; all variables log-transformed). These results reflect a tighter coupling between bacterioplankton communities and TRC concentrations than between either of these factors and nutrients in UKB.

## CyanoHAB-related seasonal conditions and water quality

Cell abundances of cyanoHAB species (cyanoHAB abundance; cells mL⁻¹) peaked in UKL, its outflow, and the reservoir sites during the bloom period (S3A Fig), and as with bacterioplankton absolute abundance, was positively correlated with bacimethrin concentrations (bolded brackets in S3A Fig next to each cyanoHAB genus). Linear regression results

also showed that cyanoHAB abundance significantly predicted levels of bacimethrin and its non-toxic analog HMP (Fig 2B,C). This result provided evidence that microbial community states characterized by a dominance of cyano-HABs were positively correlated with concentrations of HMP and bacimethrin. The relatively low adjusted $R^2$ values associated with these linear regression results could be explained by the extreme number of individual microbes that exchange each TRC at any given time, which challenges the ability for any one factor to explain a large portion of the variance in TRC concentrations. The most abundant cyanoHAB species were filamentous *Dolichospermum* and *Aphanizomenon* NIES81 spp., and *Dolichospermum* spp. displayed cell abundances as high as $1 \times 10^6$ cells mL$^{-1}$ in the Gerber Reservoir (S3A Fig).

During the pre-bloom period in May when cyanoHAB abundance was low, USGS data showed evidence for extensive hydrological mixing between UKL and the Williamson and Sprague Rivers, which flow into UKL, based on the high observed tributary discharge and lake surface elevation (S4A, S4B Fig) at this time. The Link River (UKL outflow) also displayed low and variable pH and DO concentrations during the pre-bloom period (Fig 1B,C), suggesting a suppression of primary production during this period relative to the bloom period in August. Though the bloom period saw high cyanoHAB abundance (S3A Fig), the probability of poor water quality conditions, based on surface elevation, was unexpectedly low during this time (~1,261.7 m; S4A Fig) given the abundance of cyanoHABs [46]. Link River pH and DO data suggested a modest increase in primary production (high pH and DO) during August sampling, following the main bloom and subsequent crash that likely occurred between late-July and early-August (Fig 1B,C) [46,65,66]. We also observed a complete lack of expression of genes responsible for cyanotoxin production in our metatranscriptomics results, including *mcyE* [42] and any Cosmos-Hub functional annotation terms that included "microcystin" across all databases, despite the high abundance of typical UKB toxin-producers: *M. aeruginosa* and *A. flos-aquae* (S3A Fig). These results suggest that the increase in cyanoHAB abundance during the bloom period did not degrade water quality.

## Connections between cyanoHAB abundance and bacterioplankton gene expression

When cyanoHAB abundance was high, gene expression was dominated by species of *Nostocaceae*, *Cyanobiaceae*, *Nanopelagicaceae*, *Microcystaceae*, and filamentous *Pseudanabaenaceae*. When cyanoHAB abundance was low, gene expression was dominated by putatively heterotrophic taxa such as *Burkholderiaceae* (two separate families: *Burkholderiaceae* and *Burkholderiaceae_B*), *Nanopelagicaceae*, and *Spirosomaceae* (S6 Fig). This difference in metatranscriptomes between states of low and high cyanoHAB abundances was confirmed with ANOSIM ($p = 0.003$). Conversely, 16S-based microbiome compositions did not significantly differ with changing cyanoHAB abundances (ANOSIM $p > 0.05$), despite the visual separation that existed in ordination space between high and low cyanoHAB bins for microbiome samples (S3B, S3C Fig). Based on PERMANOVA results, microbiome compositions instead primarily differed by sampling environment and season (environment: $p = 0.004$, F-statistic = 1.5, $R^2 = 0.18$; season: $p = 0.001$, F-statistic = 2.2, $R^2 = 0.09$). This statistical discrepancy suggests that, in contrast to the taxonomic compositions of metatranscriptomes (i.e., which taxa were actively transcribing genes), compositional variability in microbiomes was mainly attributed to habitat and seasonal changes rather than cyanoHAB abundance changes (S6 Fig). Microbiomes contained far more individual taxonomic units (14,547 total ASVs) than metatranscriptomes (171 total transcriptionally active strains) and these ever-changing microbiome populations may have responded more measurably to broad geographic and temporal factors like habitat and season. In contrast, highly transcriptionally active populations may have responded to quick environmental changes, such as abundance fluctuations in cyanoHAB species. Metatranscriptomics also measures a more ephemeral biological process: gene expression, whereas microbiomes are measured by DNA that degrades far slower in the environment than microbial mRNA. While it is probable that taxa with low gene expression rates still influence biogeochemistry – as these members of the community could become more metabolically active in response to environmental and timescale changes [67–69] – the most transcriptionally active populations in UKB bacterioplankton communities likely exerted the greatest biogeochemical influence [70,71].

Redundancy analysis results showed that cyanoHAB abundance, thiamin and pyrimidine concentrations ($p < 0.05$), and, to a lesser extent, bacimethrin concentrations ($0.10 > p\text{-value} > 0.05$; Fig 4B), predicted taxonomic compositions of metatranscriptomes. While its magnitude was low, the significance of the thiamin vector shows that thiamin concentrations still influence taxonomic composition of transcriptionally active bacterioplankton. Other variables such as sampling period (pre-bloom or bloom period), DO, and thiazole TRC concentrations displayed insignificant individual vectors ($p$-values $> 0.05$), yet were significant to the model prediction of metatranscriptome taxonomic compositions (model statistics in top-left box of Fig 4B). Importantly, average NextSeq read quality and sequencing depth (~30−80 million reads sample$^{-1}$) were both high, indicating that differences between metatranscriptomes were driven by recognized environmental factors rather than concerns regarding sequencing fidelity (S7 Fig).

LEfSe analysis identified bacterioplankton strains with enriched gene expression under high and low cyanoHAB abundance. Reference genomes for these taxa (i.e., those to which mRNA mapped; SI) were queried for thiamin cycling genes (bolded genes in Fig 3A) to infer potential susceptibility to bacimethrin-mediated allelopathy (susceptibility indicated by shapes in Fig 4A). Under high cyanoHAB bloom conditions all enriched cyanoHAB taxa, including *D. circinale*, *D. heterosporum* (annotated as *A. flos-aquae* on NCBI; SI), and *M. aeruginosa*, were putative thiamin prototrophs (biosynthesis gene hits in S2 Table, Fig 4A), and were thus supposedly resistant to bacimethrin. These putative prototrophs were also among the most transcriptionally active taxa under high bacimethrin concentrations, suggesting a selective pressure against auxotrophic populations at these sites. Other abundant prototrophic taxa under high cyanoHAB conditions included *Pseudanabaenaceae* and picocyanobacteria including *Cyanobium*, *Vulcanococcus*, and *Synechococcus* in Clear Lake, Link River, Rattlesnake Point, and Hank's Marsh (*Cyanobiaceae* family; relative abundances in Figs 4A and S6, S2 Table LEfSe results). Nearly all reference genomes for these taxa contained the HMP biosynthesis gene *thiC* (Fig 3A; S2 Table). Conditions of low cyanoHAB and bacimethrin favored HMP auxotrophic strains of *Limnohabitans* spp. (*Burkholderiaceae* family), nearly all of which were linked to reference genomes that lacked *thiC* and contained the pyrimidine

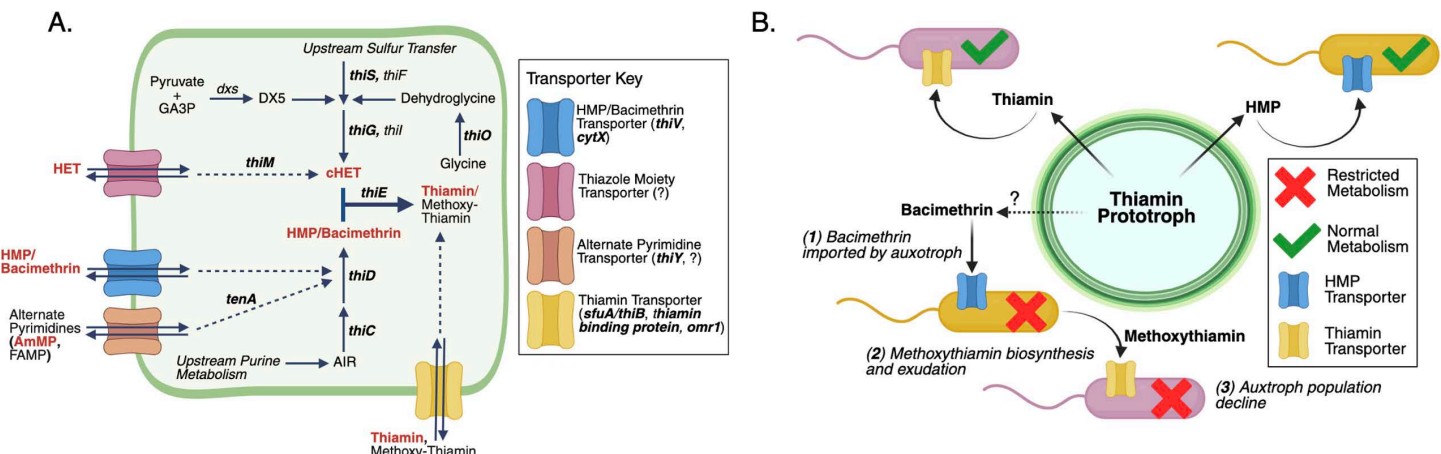

**Fig 3. Routes of thiamin biosynthesis and salvage and the hypothesized impact of thiamin antagonists on UKB surface water bacterioplankton communities. (A)** A generic cell (outlined in green) that contains routes of thiamin biosynthesis (solid black arrows) and salvage (dashed black arrows) annotated in metatranscriptomes and reference genomes. The phosphorylation of TRCs and pathway steps were left out for simplicity; see Jurgenson, Begley (75) for a more detailed diagram of thiamin biosynthesis. Red bolded labels indicate TRCs that were measured in this study and black bolded genes indicate those that were measured by hidden Markov models in reference genomes (see SI for details). Question marks indicate a lack of transporter annotations in reference databases. *"dxs"* = 1-deoxy-D-xylulose-5-phosphate synthase; "AIR" = 5-aminoimidazole ribonucleotide; "GA3P" = glyceraldehyde-3-phosphate. **(B)** A conceptual diagram showing the hypothetical metabolic impact of bacimethrin and methoxy-thiamin on thiamin auxotrophs. The question mark indicates that the source of bacimethrin, whether from prototrophic bacterioplankton or an unidentified environmental source, is unknown, but would nevertheless impact bacterioplankton that import thiamin antagonists.

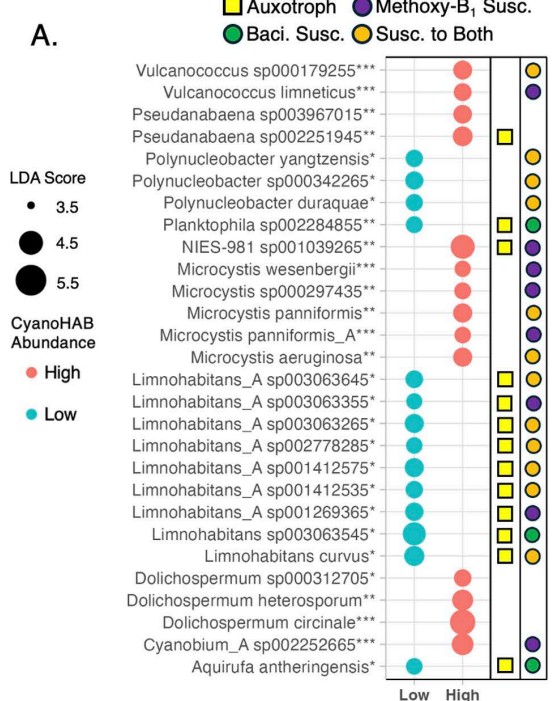

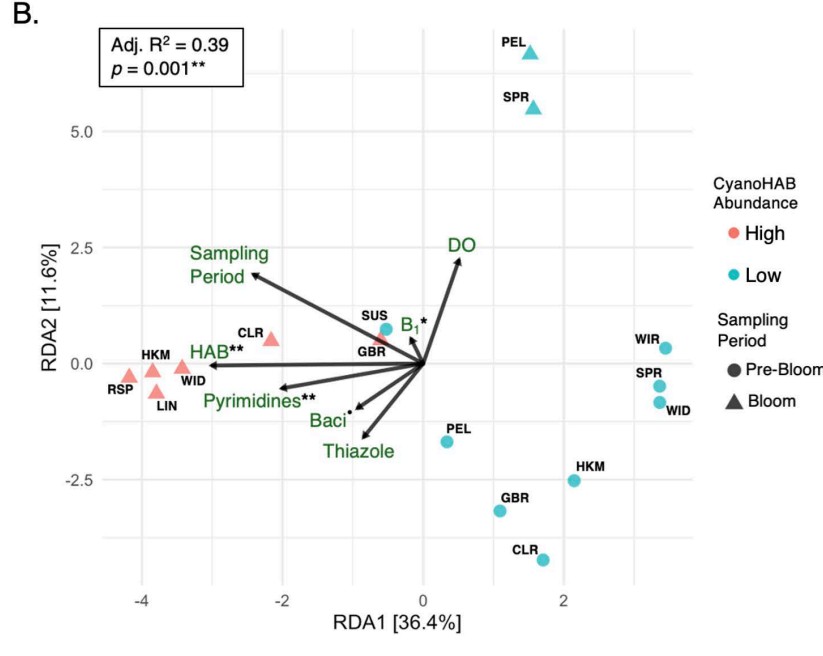

**Fig 4. CyanoHABs and TRCs influence the gene expression of UKB bacterioplankton. (A)** A bubble plot of LEfSe results that tested which strains (mRNA taxonomic annotations) most significantly differentiated high and low cyanoHAB abundance bins based on linear discriminant analysis (LDA) scores and their significance ($p < 0.05$\*, $p < 0.01$\*\*, $p < 0.001$\*\*\*; attributed to each strain). Bubble size corresponds to the magnitude of LDA scores, where LDA magnitudes increase linearly with the degree of enrichment of each strain in its respective cyanoHAB abundance bin (or how predictive each strain is of that bin), though all taxa are significantly enriched in low or high statistical bins. Yellow squares show which strains are putative thiamin auxotrophs based on their reference genomes and the absence of a yellow square indicates thiamin prototrophs. Colored circles represent bacimethrin and methoxy-thiamin susceptibility based on the presence of HMP and thiamin transporters in reference genomes, respectively. **(B)** Constrained principal component analysis (RDA; also known as redundancy analysis) displaying differences (based on Aitchison distance) between strain-level taxonomic compositions of gene transcripts and explanatory variables that significantly (based on RDA model results in top-left box) predict taxonomic compositions of transcripts. Individual $p$-values of explanatory variables are shown next to each vector name ($p < 0.1$˙, $p < 0.05$\*, $p < 0.01$\*\*). All TRC concentrations were log-transformed. "HAB" = indicator variable (presence/absence) for high cyanoHAB abundance; "DO" = dissolved oxygen concentration, "Pyrimidines" = HMP + AmMP concentrations, "Thiazole" = cHET + HET concentrations, "Baci" = bacimethrin concentrations, and "B$_1$" = thiamin concentrations.

TRC transporter genes *thiV*, *thiY*, and/or *cytX* [7] (Fig 4A, S2 Table). Other taxa favored under low cyanoHAB conditions included *Polynucleobacteria* and Planktophilia spp. whose reference genomes showed varying levels of auxotrophy and susceptibility to bacimethrin (Fig 3B).

## CyanoHAB abundance and thiamin biosynthesis gene expression

The expression of thiamin biosynthesis genes was clearly elevated when cyanoHAB abundance was high, including families of genes coding for thiazole and HMP biosynthesis (Fig 5A) and individual biosynthesis genes including *thiC*, *thiF*, and *dsx* (bolded genes in Fig 5B). This pattern was particularly strong at Rattlesnake Point during the bloom period ("RSP_B" in Fig 5B; S3 Table). This site also contained high relative abundances of the transcripts of non-filamentous and mainly prototrophic *Cyanobiaceae*, *Pseudanabaenaceae*, *Microcystaceae* (S4 Fig; see prototrophic strains in S2 Table) and low concentrations of HMP and bacimethrin (Fig 2A), indicating a net depletion of pyrimidine TRCs concomitant with thiamin biosynthesis, which is a process that has been hypothesized to occur in coastal marine environments [72]. In Gerber Reservoir, a single strain of the filamentous cyanoHAB taxon, *D. circinale* represented >99% of the bacterioplankton

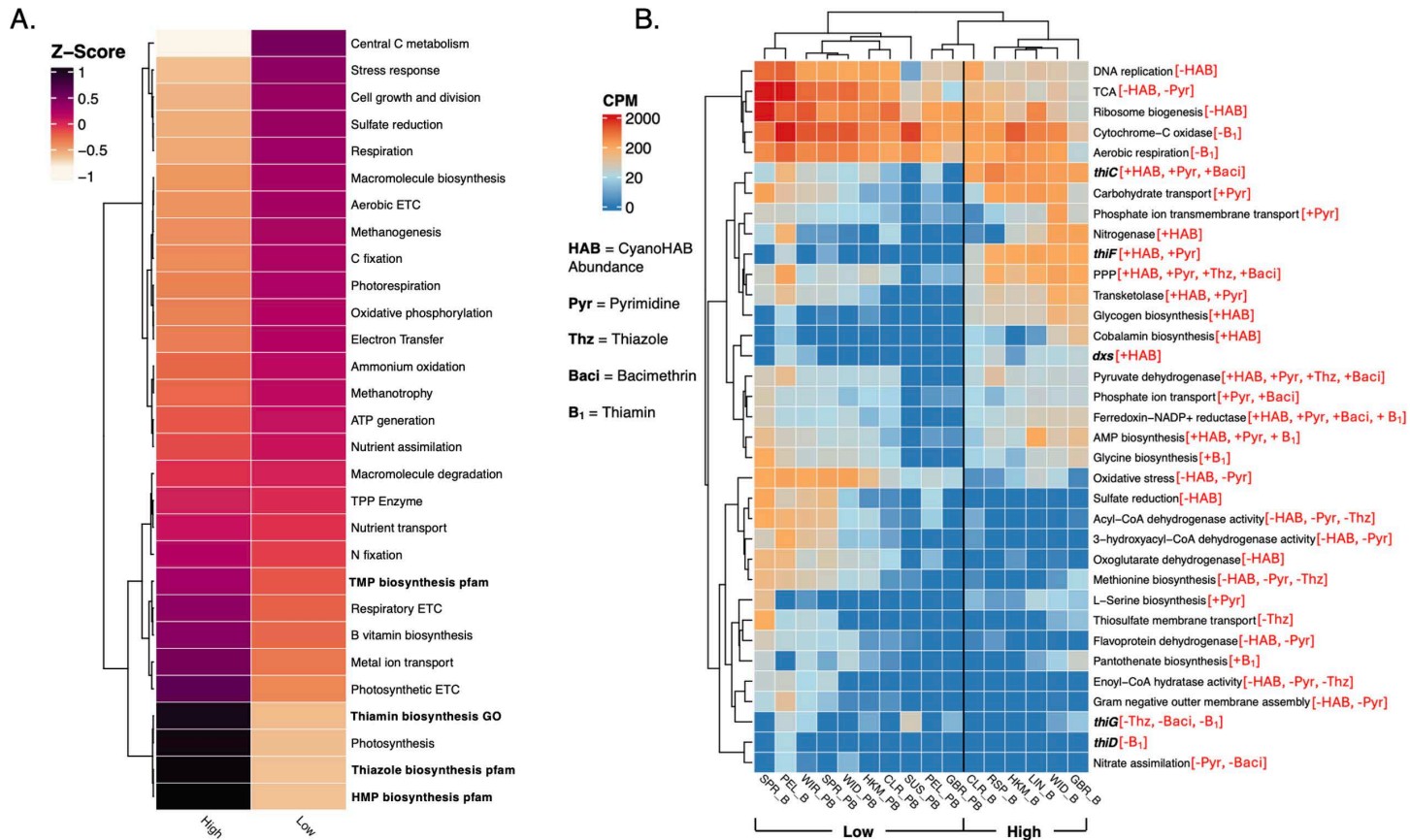

**Fig 5. Bacterioplankton gene expression changes with differences in CyanoHAB abundance and influences concentrations of TRCs. (A)** A heatmap of high-level functions (exact functional annotations displayed in panel B) and their associated z-scores averaged across all samples in each CyanoHAB biomass bin (high and low). Functions are ordered based on Euclidean distances of z-scores averaged across samples in each cyano-HAB abundance bin. **(B)** A heatmap of relative abundances (copies per million; CPM) for the top 35 most highly significant (lowest adjusted *p*-values) transcripts that correlated with cyanoHAB spp. abundances (cells mL$^{-1}$) and TRC concentrations. Samples and functional annotations are ordered based on Bray-Curtis dissimilarity. Thiamin biosynthesis genes are bolded and were annotated with the pfam database; all other transcripts were annotated with the GO terms database. Significant correlations and the direction of correlations are shown in red bracketed text (see label key). The calculation of correlations (Spearman correlations between Aitchison distances of functional annotations and variables shown in figure key; see SI) were taken from the S12 Fig correlogram. "PB" = pre-bloom and "B" = bloom. "PPP" = pentose phosphate pathway; "TCA" = tricarboxylic acid cycle; "Pyr" = correlated with at least one pyrimidine TRC: HMP and AmMP; "Thz" = correlated with at least one thiazole TRC: cHET and HET.

gene expression during the bloom period (S5 Fig). This site also displayed a significant upregulation in the expression of thiamin and other B vitamin (cobalamin and pantothenate – vitamins B$_{12}$ and B$_5$, respectively; Fig 5B and S3 Table significant functions in associated sample sites) biosynthesis genes, implicating *D. circinale* as a major microbial source for B vitamins. Together, these data suggest that prototrophic cyanobacteria in UKB, including cyanoHAB species and non-HAB-forming cyanobacteria, could produce the bulk of bacterioplankton-derived TRCs and could be B vitamin sources for higher trophic levels in the basin.

Due to incomplete or ambiguous annotations, the expression of bacimethrin biosynthesis genes was not detected in our metatranscriptomes. Many candidate genes, such as thymidylate synthase, present in both *Clostridium botulinum* and *Microcystis* spp. [20,27], have broad biochemical functions (DNA synthesis [73]), limiting their utility as diagnostic markers. Thiaminase I, found in *C. botulinum* [20] but not in queried *Microcystis* spp. reference genomes, was also absent from all queried reference genomes. Other putative bacimethrin biosynthesis genes [20,27] were similarly non-specific or absent,

precluding the confident detection of bacimethrin production via metatranscriptomics. However, bacimethrin concentrations were positively correlated with cyanoHAB abundance (Figs 2C, S3A, 4B), providing strong evidence for a microbial origin such as prototrophic taxa active under high-cyanoHAB conditions.

## Correlations between patterns of gene expression and cyanoHAB-related TRC availability

Distinct gene expression patterns under high and low cyanoHAB abundances indicate that cyanoHABs substantially influence the transcriptional activity of UKB bacterioplankton. Under high cyanoHAB abundance, there was elevated transcription of photosynthetic light reactions, the pentose phosphate pathway, the rubisco shunt, glycogen biosynthesis, gluconeogenesis, carbohydrate transport, and thiamin biosynthesis (Figs 5A, 5B, S10, S12; see S4 Table for how gene functions were categorized). These pathways reflect elevated photoautotrophic activity and *de novo* thiamin production. In contrast, samples of low cyanoHAB conditions featured enhanced transcription of genes for aerobic respiration, central carbon metabolism (tricarboxylic acid, glyoxylate, and Calvin cycles), and a range of biogeochemical processes including ammonium oxidation, methanogenesis, methanotrophy, and assimilatory sulfate reduction (Figs 5, S9–S13; S2, S3 Tables). Genes associated with cellular growth and division, such as ribosome, membrane, and cell wall biosynthesis and DNA replication pathways, also showed elevated expression in low cyanoHAB samples. The expression of fatty acid β-oxidation genes (e.g., acyl-CoA dehydrogenase) were negatively correlated with TRC concentrations and elevated under low cyanoHAB conditions (Figs 5B and S11). Oxidative stress gene expression was also highest in low-cyanoHAB samples and negatively correlated with cyanoHAB abundance and AmMP (Figs 5B, S11), suggesting that redox stress may correlate with constrained thiamin biosynthesis and availability.

Several thiamin-dependent gene pathways also varied with cyanoHAB conditions. Genes for the nitrogen fixation pathway, which can involve thiamin-dependent pyruvate ferredoxin oxidoreductase in cyanobacteria [74], showed only a modest increase under high cyanoHAB abundance (Fig 5A) even with high gene expression from filamentous and diazotrophic cyanoHAB taxa (Fig 4A). While nitrogenase gene expression was positively correlated with cyanoHAB abundance (Fig 5B), it did not correlate with TRC concentrations, suggesting a limited link between nitrogen fixation and TRC dynamics in UKB under our sampling conditions. In contrast, expression of genes related to thiamin-requiring enzymes such as pyruvate dehydrogenase (tricarboxylic acid cycle) and transketolase (pentose phosphate pathway) positively correlated with cyanoHAB abundances and pyrimidine TRCs (Fig 5B). Thiamin-dependent 1-deoxy-D-xylulose-5-phosphate synthase (*dxs*; Fig 3A), which also supports cHET biosynthesis (Fig 3A), was similarly enriched under high cyanoHAB conditions (Figs 5A and S11) [75]. It is important to note, however, that 1-deoxy-D-xylulose-5-phosphate is also a precursor for isoprenoids and pyridoxal biosynthesis, in addition to cHET, yet, regardless of the end product, *dxs* requires thiamin to function [76].

The expression of individual genes involved in HMP and thiazole biosynthesis showed opposing correlations with TRC concentrations, indicating that UKB bacterioplankton communities may uniquely cycle thiamin precursors depending on environmental conditions (Fig 5B red brackets; Spearman correlations in S11 Fig). For example, *thiC* (HMP biosynthesis) and *thiF* (cHET biosynthesis) were positively correlated with TRC concentrations while *thiG* (cHET biosynthesis pathway) and *thiD* (HMP biosynthesis or salvage) were negatively correlated with TRCs (Figs 3A and 5B). Notably, *thiG* was far more prevalent across cyanoHAB-associated reference genomes than *thiC*, suggesting a higher occurrence of pyrimidine auxotrophy than thiazole auxotrophy in UKB bacterioplankton, consistent with findings from other aquatic systems [7]. Additionally, *thiG* was highly expressed in the Sucker Springs groundwater site (Fig 5B) where the expression of other thiamin biosynthesis genes was absent, implying that thiazole biosynthesis may be a more conserved trait across diverse UKB bacterioplankton and potentially explaining its weak correlation with cyanoHAB abundance.

## A hypothetical model relating thiamin usage with hypereutrophic biogeochemistry

We developed a conceptual model (Fig 6) based on our metatranscriptomics data to illustrate how bacterioplankton could use TRCs under different cyanoHAB conditions. Gene pathways with expressions that correlated with cyanoHAB abundance, TRC concentrations (see red brackets in Fig 5B), and putative bacimethrin allelopathic effects were integrated into

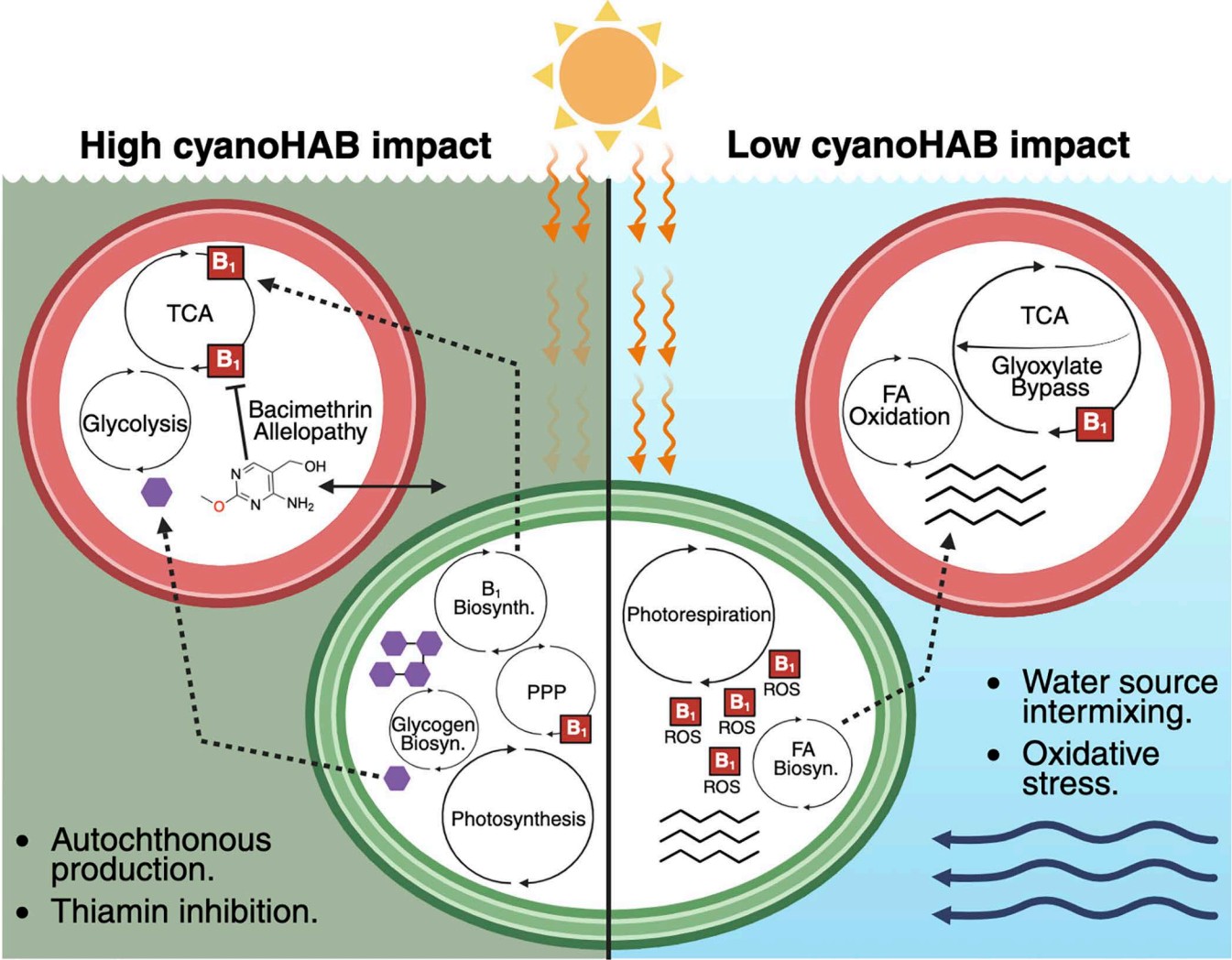

**Fig 6. The impact of cyanoHABs correlates with cellular biochemical changes of bacterioplankton that influence how thiamin is used by cellular enzymes.** The left and right sides of the figure (either side of the middle green cell, separated by the solid black line) display biochemical patterns of bacterioplankton gene expression in generic UKL habitats that, aside from the association between thiamin and ROS, are enriched in either high or low cyanoHAB abundance bins, which are based on Cosmos-Hub LEfSe or differential expression (based on z-scores) results. Dark green represents murkier surface water and poor sunlight penetration from cyanoHAB scums and light blue represents the opposite. The two panels do not portray the presence/absence of pathways, but rather the relative enrichment of gene expression patterns that were upregulated in each scenario. Red "$B_1$" squares indicate metabolic pathways that contain thiamin-requiring enzymes or, in the case with reactive oxygen species (ROS), the use of thiamin to degrade ROS. Bacimethrin allelopathy is indicated by the exchange of bacimethrin – with its chemical structure shown – with the exogenous environment. Green and red cells portray generic photoautotrophs and heterotrophs, respectively. Purple hexagons portray glucose and represent autochthonous dissolved organic matter exuded by the photoautotroph. The bottom right panel horizontal arrows representing water current indicate greater input and mixing of multiple water sources such as groundwater and tributary water mixing with lake water or reservoir water. PPP = pentose phosphate pathway; TCA = tricarboxylic acid cycle; FA = fatty acid; Thiamin inhibition = thiamin-requiring enzyme inhibition from bacimethrin allelopathy.

cellular models representing the metabolic profiles of bacterioplankton under conditions of high and low cyanoHAB impact (Fig 6). While photosynthesis genes were expressed at all sites (S13 Fig), differences were shown in ATP-expending pathways: photorespiration was upregulated under low cyanoHAB conditions, particularly at Pelican Bay (S3 Table) where high groundwater input can buffer against cyanoHAB-driven water quality degradation [41,77].

Sites with low cyanoHAB impact, such as Pelican Bay, also showed elevated expression of gene pathways involved in reactive oxygen species (ROS) mitigation (Figs 5 and 6), which could have been driven by increased light stress and oxidative pressure in the absence of shading by cyanoHAB surface scums [78–80]. Conversely, sites with high cyanoHAB abundance could have experienced reduced light penetration due to the formation of surface scums [81,82], further influencing photoautotrophic dynamics (Fig 6). Collectively, these observations show how thiamin metabolism, oxidative stress response, and light-driven energy allocation could differ across cyanoHAB gradients. This synthesis highlights the broader ecological consequences of cyanoHABs for microbial interactions and (micro)nutrient cycling in eutrophic systems.

## Discussion

Our data support our hypothesis that cyanoHAB abundance was correlated with dissolved TRC concentrations and cyanoHAB species influenced patterns of bacterioplankton gene expression. Additionally, the observation that bacimethrin was present in all samples, with the highest concentrations (ca. 5–25 pM) measured under conditions of high cyanoHAB abundance (Fig 2), suggests that, based on these correlative results, bacimethrin allelopathy could be maximized in the presence of cyanoHABs in UKB surface waters. Additionally, we observed that dissolved concentrations of bacimethrin and its non-toxic analog HMP were positively correlated with cyanoHAB species abundances and were in similar concentrations across diverse environments and cyanoHAB-related seasons, further indicating that the distributions of these sister-compounds are controlled by the same mechanism (Fig 2). This finding is not surprising because the known mechanism of action for bacimethrin-based allelopathy is competitive inhibition of HMP binding transporters and enzymes. Thus, we speculate that microbial uptake of both compounds via the same high affinity HMP transporters (such as ThiV and CytX) is the specific process controlling observed dissolved concentrations of bacimethrin and HMP.

A study using cell cultures of the ubiquitous marine bacterium *Candidatus* Pelagibacter st. HTCC7211, a known HMP auxotroph, reported half saturation constants (Km) for the HMP transporter ThiV to be within the range of observed bacimethrin and HMP concentrations in UKB (Km, 9.5 pM to 1.2nM; Fig 2) [83]. The similarity between the HTCC7211 Km and substrate concentrations found in the UKB provide strong evidence that the lower bounds of uptake affinity were governing dissolved concentrations in the environment. While previous research [83] shows that HMP transporters are highly specific and not subject to competitive inhibition, bacimethrin was not evaluated in these experiments and no HMP uptake kinetics studies have been conducted on freshwater microbial species. Further, we observed that genes for high affinity HMP transporters were present across reference genomes (pyrimidine salvage gene column in S2 Table), especially in taxa whose transcriptional activity was negatively correlated with cyanoHAB abundance – e.g., *Limnohabitans* spp. (Fig 4A).

The low observed concentrations of AmMP, HMP, and bacimethrin relative to thiamin and thiazole (Fig 2A) may reflect high cellular requirements for, and thus rapid uptake of, pyrimidine TRCs by HMP auxotrophs. These low concentrations were observed in parallel with high transcriptional activity of the HMP biosynthesis gene *thiC* (Fig 5B); this could indicate that recently synthesized pyrimidine TRCs are more rapidly removed from the dissolved pool than other less essential TRCs, which would accumulate to higher concentrations without such rapid removal. This high microbial demand for pyrimidine congeners is supported by past research [7,11] and could increase the allelopathic impact of bacimethrin (Fig 3A). CyanoHAB species abundances and gene expression levels strongly correlated with pyrimidine (bacimethrin, HMP, and AmMP) TRC availability, evidenced by linear regression, correlational, and gene expression results that showed increasing HMP and bacimethrin concentrations and *thiC* expression with high cyanoHAB abundance (Figs 2B,C and 5B). These results further reinforce the tight connections between cyanoHABs, pyrimidine TRCs, and bacimethrin allelopathy in UKB.

The recent discovery that *Microcystis* spp., which were highly transcriptionally active with high cyanoHAB abundance in our dataset (Figs 4 and S6), can produce bacimethrin in culture [27] suggests these cyanoHAB species may be important sources of bacimethrin in UKB. *Microcystis* spp. use many strategies to rapidly respond to changing environmental

conditions and outcompete other bacterioplankton, which is exemplified by their expansive pan-genome [29] – the genetic repertoire of survival strategies – including microcystin production, high-affinity nutrient transporters, and efficient buoyancy mechanisms [84,85]. Reference genomes of *Microcystis* spp., which represent transcriptionally active strains in UKB, also displayed the highest variety of thiamin biosynthesis and salvage genes across all queried genomes (S2 Table), indicating the potential for both full *de novo* thiamin biosynthesis and salvage of thiamin and pyrimidine TRCs (Fig 3A). Considering the adaptability of these taxa to the environment, the regulation of *Microcystis spp.* thiamin-cycling genes could be controlled by thiamin-dependent riboswitches [86], potentially allowing *Microcystis* spp. to switch between biosynthesis and salvage based on TRC availability. The ability of transcriptionally-regulated *de novo* thiamin biosynthesis could also enable *Microcystis* spp. to avoid importing toxic bacimethrin. Bacimethrin production may be yet another mechanism used by *Microcystis* spp., and potentially other cyanoHAB species, to dominate eutrophic watersheds.

Our data indicate that the influence of TRCs on bacterioplankton gene expression depends on the relative abundances of active thiamin prototrophs and auxotrophs in each sample. Although both thiazole and pyrimidine TRCs are essential and can be salvaged by cells, pyrimidines showed the strongest relationship with total transcriptionally active community compositions in this study (Fig 4B). This supports the idea that the specific type of thiamin auxotrophy – determined by the presence or absence of biosynthesis and salvage genes in reference genomes (Fig 3A) – influences which TRCs are most essential to a given microbial community. Our RDA results further support this by showing that pyrimidine TRCs and thiamin had a stronger influence on bacterioplankton gene expression than thiazole TRCs (TRC vectors in Fig 4B). This may reflect the near absence of thiazole auxotrophs among reference genomes, most of which showed a capacity for thiazole biosynthesis and lacked the salvage gene, *thiM* (Fig 3A), including those correlated with cyanoHABs (LEfSe results in S2 Table). Our results suggest that thiazole TRCs were less essential and thus exerted weaker influence on community structure than pyrimidine TRCs in UKB. This underscores the likely importance of bacimethrin allelopathy in the UKB system, as it would target HMP auxotrophs. Future bacimethrin-based microbial ecophysiology research (similar to [27]) will be required to identify causation rather than correlation because other factors in addition to TRCs, such as nutrient limitation and variabilities in redox conditions, unquestionably influence microbial community dynamics.

UKL environmental conditions reflected how different lake habitats, influenced by unique seasonal hydrological factors (groundwater and tributary flow) and cyanoHABs [41], can foster bacterioplankton communities that uniquely cycle TRCs. For example, all pre-bloom UKL habitats displayed low thiamin biosynthesis gene expression (Fig 5A; S3 Table) and high transcriptional activities of HMP auxotrophs (S6 Fig; S2 Table). Conversely, as the summer progressed (Figs 1B,C), we hypothesize that autochthonous TRC production by cyanobacteria in UKL increased, concurrent with a lower UKL surface elevation and tributary input that reduces dispersal of cells and TRCs into the lake (S4A, S4B Fig). These observations from UKB may parallel observations in global marine environments where diverse environmental conditions select for unique microbial communities that establish dissolved TRC availability [13,72,87]. In further support of this notion, dissolved TRCs across river habitats in the Sacramento River watershed, a system controlled by completely distinct environmental conditions from UKB, showed far lower thiamin concentrations (a median of 0.23 pM across seasons, compared to a UKB median of 40.2 pM across seasons) and relative abundances of cyanobacteria than UKB [18]. Similar to other aquatic environments adjacent to urban centers [88], the unique environmental conditions within UKB habitats influenced the composition of bacterioplankton communities (S3B, S3C Fig ordinations). This environmental selection may have also had a measurable influence on TRC concentrations and the potential for bacimethrin allelopathy, based on which bacterioplankton populations were favored by specific environmental conditions of each UKB habitat and the thiamin-cycling genes encoded in the genomes of these populations.

The relatively high bacimethrin levels associated with cyanoHABs in UKB negatively correlated with auxotrophic and globally ubiquitous heterotrophic bacterioplankton. This suggests that high bacimethrin and cyanoHAB abundance could favor prototrophs and select against auxotrophs (Figs 3B and 4A; S2 Table). While this proposed mechanism will need to be validated with direct experimental evidence, our data did show a near complete depletion of the gene expression of

auxotrophic *Limnohabitans* spp., during periods of high cyanoHAB species abundance (Figs 4 and S6; S2 Table). *Limnohabitans* (*Burkholderiaceae*) are globally ubiquitous in freshwater ecosystems and are capable of heterotrophic growth on organic exudates released by algae [89–91]. Reference genome analysis suggests that *Limnohabitans* spp. lack the *thiC* gene (S2 Table), implying a dependency on exogenous pyrimidine TRCs, which could be alleviated by pyrimidine importation via ThiV, ThiY, and CytX transporters and the TenA salvage pathway (S2 Table) [16,87,92]. The lack of transcriptional activity of these auxotrophs in several sites dominated by cyanoHAB species (S8 Fig) points towards a bacimethrin-based, or yet to be discovered, allelopathic mechanism inhibiting their metabolic activity (Fig 3B). This could explain why the previously observed co-occurrence of *Limnohabitans* spp. with other photoautotrophs [90] was not observed in UKB habitats.

The expression of biogeochemically relevant genes related to carbon-cycling, energy acquisition, and cell stress across sites with varying cyanoHAB impact revealed additional metabolic pathways potentially influenced by TRCs and cyanoHAB abundances. These pathways included genes for fatty acid degradation enzymes, the glyoxylate cycle (or "bypass"), and oxidative stress response pathways (Figs 5, 6, and S11), all of which are connected to cellular thiamin demand. For example, elevated expression of fatty acid degradation genes in low cyanoHAB samples could reduce reliance on the thiamin-dependent pyruvate dehydrogenase, as fatty acids can directly enter the TCA cycle (Fig 6) [93]. Similarly, the glyoxylate bypass circumvents the need for thiamin-requiring 2-oxoglutarate dehydrogenase (Fig 6) and glyoxylate bypass gene expression was negatively correlated with cyanoHAB abundance (S11 Fig). Periods of intensified oxidative stress experienced by UKB bacterioplankton may further deplete intracellular thiamin due to the thiazole ring's role in deactivating ROS [94] (Fig 6). Enrichment of photorespiration gene expression in low cyanoHAB sites provides additional evidence of oxidative stress and possible thiamin limitation (Fig 6). This energy-consuming process is typically associated with oxygen-rich conditions and has been proposed as a protective mechanism against photo-oxidative damage in cyanobacterial mats [95,96]. Our observed negative correlation between photorespiration gene expression and dissolved thiamin concentrations (S11 Fig), along with increased expression of these genes under low cyanoHAB conditions (Fig 5B and S2 Table), support a link between photorespiration, cell stress, and limited environmental thiamin availability in UKB sites (Fig 6).

Using multiple methods, we find that transcriptionally active UKB bacterioplankton communities influence TRC availability and could be uniquely susceptible to an allelopathic thiamin antagonist. Each thiamin-requiring metabolic pathway (see Fig 6) actively expressed by UKB bacterioplankton has the potential to be competitively inhibited by bacimethrin. As a result, changes in the expression of these pathways resulting from bacimethrin allelopathy could exert strong biogeochemical influences on UKB surface waters and the persistence of cyanoHABs.

## Conclusion

CyanoHABs are a nuisance to freshwater ecosystems across the planet [97]. Our results point to the previously unrecognized importance of a toxic HMP analog, bacimethrin, as an allelopathic factor that could promote cyanoHAB dominance and persistence. Microbial HMP auxotrophy is common, so, while our results are specific to UKB, we expect that similar ecosystem-scale impacts occur in other watersheds similarly plagued by bloom-forming *Microcystis*, *Dolichosperum*, and *Aphanizomenon* species. While future research showing causation must be conducted to confirm its impact, bacimethrin allelopathy could be yet another mechanism for niche expansion exploited by cyanoHAB species, allowing these organisms to gain and maintain dominance in eutrophic watersheds.

## Supporting information

**S1 Methods. Additional methodological detail is provided in the Supplemental Information.**
(DOCX)

**S1 Fig. Nutrient ratios change based on the sampling season.** µM nutrient ratios of DIN:phosphate (P) across sample sites in the pre-bloom and bloom time periods. Dashed lines are drawn at 16 µM to indicate the Redfield ratio of N:P of 16:1.
(TIF)

**S2 Fig. Absolute abundance of 16S copies mL⁻¹ based on *T. thermophilus* spike-in internal standard results.** Shapes correspond to sampling time periods; "Pre-Bloom" = May 2023, "Bloom" = August 2023. Lines are placed in between sample sites in color key to distinguish sampling environments (same order as shown in panel A x-axis).
(TIF)

**S3 Fig. CyanoHAB cellular abundances (cells mL⁻¹; abundance) are highest during the bloom time period and influence bacterioplankton.** (A) Heatmap showing the abundance of each unique cyanoHAB genus based on 16S gene absolute abundances and copy numbers (see Methods). Samples taken during the bloom period (colored green) are shown first followed by pre-bloom (colored orange) samples and cyanoHAB abundance bins are indicated above site names. Two-sided Spearman correlations and significance levels *(p < 0.1˙, p < 0.05\*, p < 0.01\*\*)* are shown in bolded brackets next to each cyanoHAB genus and represent correlations between the total biomass of all ASVs in each of the four unique genera and bacimethrin concentrations. PCoA, or metric multidimensional scaling (MDS), ordinations were constructed to display differences (based on Aitchison distance) between (B) ASV-level 16S-based microbial community compositions and (C) strain-level taxonomic compositions of gene transcripts across sites. Shapes and colors represent sampling environments ("Location") and cyanoHAB abundance bins, respectively.
(TIF)

**S4 Fig. Water quality related factors change in UKL across the summer.** (A) Surface water elevation and (B) tributary discharge (Sprague and Williamson Rivers) were plotted from USGS hourly water quality monitoring stations between May 5th and Sept. 1st, 2025.
(TIF)

**S5 Fig. Stacked bar plots showing bacterioplankton families of the highest relative abundances based on 16S) and (B mRNA taxonomic annotations.** Samples are binned by cyanoHAB abundance. "PB" = pre-bloom and "B" = bloom.
(TIF)

**S6 Fig. Stacked bar plots showing bacterioplankton families of the highest relative abundances based on mRNA taxonomic annotations.** Samples are binned by cyanoHAB abundance. "PB" = pre-bloom and "B" = bloom.
(TIF)

**S7 Fig. mRNA-seq reads displayed high read quality and depth.** Read quality reports generated from the Cosmos-Hub bioinformatics pipeline displaying the (A) Phred scores of merged forward and reverse samples and (B) read depths and the percent of reads that failed Cosmos-Hub quality control steps.
(TIF)

**S8 Fig. Unique taxa dominate each cyanoHAB abundance bin.** Relative abundances of the transcripts from the top 10 most abundant bacterioplankton genera across samples. Samples are binned by cyanoHAB abundance (high and low cohorts). This plot was generated by the Cosmos-Hub bioinformatics HUB. "PB" = pre-bloom and "B" = bloom.
(TIF)

**S9 Fig. MetaCyC pathways differ across sites and between sites of different cyanoHAB impact.** Relative abundance (copies per million) of MetaCyc pathways across sample sites binned into high (samples under pink bar) and low

(samples under teel bar) cyanoHAB abundnace (or impact). This plot was generated by the Cosmos-Hub bioinformatics HUB. "PB"=pre-bloom and "B"=bloom.
(TIF)

**S10 Fig. MetaCyC pathways differ across sites and between sites of different cyanoHAB impact.** Relative abundance (copies per million) of MetaCyc pathways across sample sites binned into high (samples under pink bar) and low (samples under teal bar) cyanoHAB abundance. This plot was generated by the Cosmos-Hub bioinformatics HUB. "PB"=pre-bloom and "B"=bloom.
(TIF)

**S11 Fig. Bacterioplankton gene expression that were differentially enriched in cyanoHAB abundance bins also correlated with cyanoHAB abundances (continuous values; cells ml$^{-1}$) and TRC concentrations.** Correlogram of GO term and pfam annotations whose CLR-transformed relative abundances (copies per million) significantly correlate (adjusted Spearman $p < 0.05$) with at least one thiamin congener, bacimethrin, and/or cyanoHAB abundance. "+B"=significant lefse enrichment ($p < 0.05$) in high CyanoHAB biomass bin and "-B"=significant lefse enrichment ($p < 0.05$) in low cyanoHAB abundance bin, respectively, based on annotations from at least one Cosmos-Hub functional reference database. Threonine=threonine synthase activity; sulfate reduction=assimilatory sulfate reduction; RPP=reductive pentose phosphate pathway; ETC=electron transport chain; PPP=pentose phosphate pathway. Larger asterisks indicate the most significant correlations (adjusted $p < 0.01$).
(TIF)

**S12 Fig. GO term-based functional annotations differ based on cyanoHAB abundance.** All significant ($p < 0.05$) GO terms (biological processes) that were enriched in cyanoHAB abundance bins (high and low) based on lefse. The x-axis displays linear discriminant analysis scores. This plot was generated by the Cosmos-Hub bioinformatics HUB.
(TIF)

**S13 Fig. All manually curated (see Supp.** Methods) functional annotations differ across sample sites. A heatmap of relative abundances (copies per million; CPM) of all the manually curated GO terms and pfam annotations across samples. Sample and function seriation (based on hierarchical clustering) was performed based on Bray-Curtis dissimilarity. "PB"=pre-bloom and "B"=bloom.
(TIF)

**S1 Table. Mean TRC concentration and standard deviation values derived from three technical replicates per sample.** Months of sampling and environments are indicated.
(XLSX)

**S2 Table. Reference genomes and their associated TRC-cycling genes taken from taxonomically annotated transcripts that were significantly enriched or depleted with cyanoHAB abundance, based on Cosmos-Hub LEfSe results.** See description of supplemental tables for more details.
(XLSX)

**S3 Table. Functional pathways that, based on z-score results (see main text Methods), were significantly expressed with high or low cyanoHAB abundance and their associated p- and adjusted p-values.**
(XLSX)

**S4 Table. The hierarchical organization of functional pathways of interest based on GO terms and pfam annotations performed by the Cosmos-Hub.**
(XLSX)

## Acknowledgments

We thank the USFWS Klamath Falls Office including Rodger Gwiazdowski, Charlee Cramer, Ronald Twibell, and Christinia Kruse and U.S. Bureau of Reclamation staff Brock Phillips for their assistance with field sampling and project development. We thank Stephen Giovannoni for his support, mentorship, and use of his laboratory space. We thank Jeff Morré and Jaewoo Choi for assistance with mass spectrometry. We thank Beth Ahner for her advice, support, and collaboration surrounding bacimethrin. Finally, we thank the broader community of Salmonid Thiamin Deficiency Complex scientists for helpful and inspirational conversations that have shaped the direction of this research including workshops at the National Center for Ecological Analysis and Synthesis and the USGS John Wesley Powell Center for Analysis and Synthesis.

## Author contributions

**Conceptualization:** Kelly C. Shannon, Frederick S. Colwell, Byron C. Crump, Christie Nichols, Clifford E. Kraft, Christopher P. Suffridge.

**Data curation:** Kelly C. Shannon, Christopher P. Suffridge.

**Formal analysis:** Kelly C. Shannon, Clifford E. Kraft, Christopher P. Suffridge.

**Funding acquisition:** Kelly C. Shannon, Christie Nichols, Christopher P. Suffridge.

**Investigation:** Kelly C. Shannon, Frederick S. Colwell, Elizabeth Brennan, Gillian St. John, Robin Gould, Christopher Hartzell, McKenzie Wasley, Christie Nichols, Clifford E. Kraft, Christopher P. Suffridge.

**Methodology:** Kelly C. Shannon, Frederick S. Colwell, Byron C. Crump, Elizabeth Brennan, Gillian St. John, Robin Gould, Christopher Hartzell, McKenzie Wasley, Christie Nichols, Christopher P. Suffridge.

**Project administration:** Kelly C. Shannon, Byron C. Crump, Christopher P. Suffridge.

**Resources:** Kelly C. Shannon, Frederick S. Colwell, Byron C. Crump, McKenzie Wasley, Christie Nichols, Christopher P. Suffridge.

**Software:** Kelly C. Shannon, Christopher P. Suffridge.

**Supervision:** Kelly C. Shannon, Frederick S. Colwell, Christopher P. Suffridge.

**Validation:** Kelly C. Shannon, Christopher P. Suffridge.

**Visualization:** Kelly C. Shannon, Byron C. Crump, Christopher P. Suffridge.

**Writing – original draft:** Kelly C. Shannon, Frederick S. Colwell, Christopher P. Suffridge.

**Writing – review & editing:** Kelly C. Shannon, Frederick S. Colwell, Byron C. Crump, McKenzie Wasley, Christie Nichols, Clifford E. Kraft, Christopher P. Suffridge.

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
