## [Decision Letter · Decision Letter 0]

5 Sep 2025

PONE-D-25-38532The allelopathic vitamin B1 antagonist bacimethrin impacts microbial gene expression in a hypereutrophic watershed dominated by cyanobacterial bloomsPLOS ONE

Dear Dr. Suffridge,

Thank you for submitting your manuscript to PLOS ONE. After careful consideration, we feel that it has merit but does not fully meet PLOS ONE’s publication criteria as it currently stands. Therefore, we invite you to submit a revised version of the manuscript that addresses the points raised during the review process.

We look forward to receiving your revised manuscript.

Kind regards,

Barathan Balaji Prasath

Academic Editor

PLOS ONE

Journal Requirements:

2. Thank you for stating the following financial disclosure: [This work was funded by United States Fish and Wildlife Service grant F22AC01810-01 and California Department of Fish and Wildlife grant Q2196012, both to Christopher P. Suffridge. Additional personnel funding for Christopher P. Suffridge was provided by National Science Foundation grant DEB-1639033. Mass spectrometry instrumentation at the OSU Mass Spectrometry Center was supported by National Institutes of Health grant 1S10RR022589-1.]

3. Thank you for stating the following in the Acknowledgments Section of your manuscript: [We thank the USFWS Klamath Falls Office including Rodger Gwiazdowski, Charlee Cramer, Ronald Twibell, and Christinia Kruse and U.S. Bureau of Reclamation staff Brock Phillips for their assistance with field sampling and project development. We thank Stephen Giovannoni for his support, mentorship, and use of his laboratory space. We thank Jeff Morré and Jaewoo Choi for assistance with mass spectrometry. Finally, we thank the broader community of Salmonid Thiamin Deficiency Complex scientists for helpful and inspirational conversations that have shaped the direction of this research (workshops funded by NCEAS Morpho, USGS John Wesley Powell Center for Analysis and Synthesis, and NSF). We thank Beth Ahner for her advice, support, and collaboration surrounding bacimethrin. This work was funded by United States Fish and Wildlife Service grant F22AC01810-01 and California Department of Fish and Wildlife grant Q2196012, both to Christopher P. Suffridge. Additional personnel funding for Christopher P. Suffridge was provided by National Science Foundation grant DEB-1639033 Mass spectrometry instrumentation at the OSU Mass Spectrometry Center was supported by National Institutes of Health grant 1S10RR022589-1. The authors declare no conflict of interest.]

Please remove any funding-related text from the manuscript and let us know how you would like to update your Funding Statement. Currently, your Funding Statement reads as follows: [This work was funded by United States Fish and Wildlife Service grant F22AC01810-01 and California Department of Fish and Wildlife grant Q2196012, both to Christopher P. Suffridge. Additional personnel funding for Christopher P. Suffridge was provided by National Science Foundation grant DEB-1639033. Mass spectrometry instrumentation at the OSU Mass Spectrometry Center was supported by National Institutes of Health grant 1S10RR022589-1.]

4. We noted in your submission details that a portion of your manuscript may have been presented or published elsewhere. [A small subset of the dissolved bacimethrin data (2 stations) was recently published in Yazadani et al. In that manuscript these data were used to provide evidence that bacimethrin is present in the environment. No further analysis of the data was included in that paper.

The complete bacimethrin dataset is included here to allow for complete ecological analysis. We have explained htis overlap in the methods and have cited Yazdani et al.

Mohammad Yazdani, Christopher P. Suffridge, Fangchen Liu, Cait M. Costello, Zhiyao Zhou, Gillian St. John, et al. Harmful algal bloom species Microcystis aeruginosa releases thiamin antivitamins to suppress competitors. mbio 2025. doi: https://doi.org/10.1128/mbio.01608-25.] Please clarify whether this [conference proceeding or publication] was peer-reviewed and formally published. If this work was previously peer-reviewed and published, in the cover letter please provide the reason that this work does not constitute dual publication and should be included in the current manuscript.

Additional Editor Comments:

Reviewer #1:

n the manuscript “ The allelopathic vitamin B1 antagonist bacimethrin impacts microbial gene expression in a hypereutrophic watershed dominated by cyanobacterial blooms” it is clearly indicated that increased cyanobacterial bloom abundance is associated with increased concentrations of the thiamine and its antagonist, bacimethrin, and increased expression of thiamine biosynthesis genes (thiC, thiF, and dxs). However, some minor errors and obscurities need to be clarified before publishing.

1. The year "2025" in the Figure 1B, C caption seems to be a typographical error. Because the Materials and Methods section states that “ samples were collected May 1-3 and August 28-30, 2023, during daylight”.

2. Although linear regressions show that cyanoHAB abundance significantly predicts HMP and bacimethrin concentrations. But the low adjusted R² values (0.18 for HMP and 0.38 for bacimethrin) indicate that there are also other factors controlling the levels of these two chemicals. And authors were unable to identify potential genes responsible for bacimethrin synthesis. However, previous study in M. aeruginosa has suggested that genes encoding glycosyltransferase, methyltransferase, and thiaminase I may be involved in bacimethrin production, like their homologs in the bcm gene cluster found in Clostridium botulinum.

3. Why did you set a threshold of >1,000 cells mL⁻¹ for CyanoHAB high and low?

Reviewer #2:

The study establishes bacimethrin as an allelopathic factor influencing microbial gene expression, however, the discussion does not clearly differentiate how much of the observed gene expression is bacimethrin-specific versus broader Eutrophication /cyano HAB effects. Stronger evidence separating correlation from causation is needed.

The chemical quantification of bacimethrin and thiamin should include detection limits, calibration methods, and reproducibility for LC-MS/MS measurements.

The metatranscriptomics is briefly mentioned, add details on read mapping thresholds, normalization, and statistical treatment (e.g., correction for multiple testing in LEfSe analyses).

The conclusion that bacimethrin confers a competitive advantage to prototrophs (e.g., cyanoHAB taxa) is plausible, but the manuscript does not present direct experimental evidence (e.g., controlled culture assays with bacimethrin). This should be acknowledged as a limitation.

Alternative explanations such as nutrient limitation, redox dynamics, or co-occurring toxins could also structure microbial communities. The discussion should weigh these factors.

The manuscript uses a large dataset (metabolites, community composition, transcripts), but integration sometimes feels descriptive rather than mechanistic. For example, correlations between bacimethrin and gene expression are presented, but no causal or predictive modeling is attempted.

The claim that bacimethrin is “nearly equimolar” with HMP may be misleading given the high variability and large error ranges in Table 1. Statistical support for “equimolar” is weak.

Redundancy analysis (RDA) is included, but effect sizes are low, raising questions about biological significance.

The introduction is strong and well-referenced, but could more explicitly state the hypothesis: is the expectation that bacimethrin actively suppresses auxotrophs, or that it is merely a biomarker of bloom dominance?

The discussion occasionally overstates certainty (e.g., “bacimethrin provides a competitive advantage”) without caveats.

Some pathway schematics (e.g., Fig. 6) are visually complex and may be simplified for clarity.

Supporting information tables (S1–S4) are extensive but not fully summarized in the main text.

“TRCs” (thiamin and related compounds) is introduced but should be used more consistently throughout.

The term “allelopathy” may imply direct inhibition; clarify that the evidence here is correlative.

Only 16 of 24 samples yielded usable RNA. This introduces bias, but the limitation is not fully acknowledged.

Metatranscriptome and 16S community analyses show inconsistent patterns (DNA shows environment/season as stronger drivers; RNA shows cyanoHAB-driven changes). This contradiction is under-discussed.

The presence of biosynthesis/salvage genes in reference genomes is used to infer ecological function, but no expression or enzyme activity data confirm functionality.

Some tests (e.g., Wilcoxon signed-rank, PERMANOVA) are mentioned, but exact test statistics, effect sizes, and corrections for multiple testing are not always reported.

Revise to emphasize limitations and uncertainties (lack of experimental causation, possible confounders).

Strengthen methodological transparency for chemical and transcriptomic analyses.

Clarify the central hypothesis and refine the narrative to avoid overstated conclusions.

Simplify and better integrate figures/tables with the text.

Some citations like HMP auxotrophy studies are outdated, try to add recent ones.

Reviewers' comments:

Reviewer's Responses to Questions

**Comments to the Author**

1. Is the manuscript technically sound, and do the data support the conclusions?

Reviewer #1: Partly

Reviewer #2: Partly

2. Has the statistical analysis been performed appropriately and rigorously? 

Reviewer #1: Yes

Reviewer #2: Yes

3. Have the authors made all data underlying the findings in their manuscript fully available?

Reviewer #1: Yes

Reviewer #2: Yes

4. Is the manuscript presented in an intelligible fashion and written in standard English?

Reviewer #1: Yes

Reviewer #2: Yes

5. Review Comments to the Author

Reviewer #1: In the manuscript “ The allelopathic vitamin B1 antagonist bacimethrin impacts microbial gene expression in a hypereutrophic watershed dominated by cyanobacterial blooms” it is clearly indicated that increased cyanobacterial bloom abundance is associated with increased concentrations of the thiamine and its antagonist, bacimethrin, and increased expression of thiamine biosynthesis genes (thiC, thiF, and dxs). However, some minor errors and obscurities need to be clarified before publishing.

1. The year "2025" in the Figure 1B, C caption seems to be a typographical error. Because the Materials and Methods section states that “ samples were collected May 1-3 and August 28-30, 2023, during daylight”.

2. Although linear regressions show that cyanoHAB abundance significantly predicts HMP and bacimethrin concentrations. But the low adjusted R² values (0.18 for HMP and 0.38 for bacimethrin) indicate that there are also other factors controlling the levels of these two chemicals. And authors were unable to identify potential genes responsible for bacimethrin synthesis. However, previous study in M. aeruginosa has suggested that genes encoding glycosyltransferase, methyltransferase, and thiaminase I may be involved in bacimethrin production, like their homologs in the bcm gene cluster found in Clostridium botulinum.

3. Why did you set a threshold of >1,000 cells mL⁻¹ for CyanoHAB high and low?

Reviewer #2: � The study establishes bacimethrin as an allelopathic factor influencing microbial gene expression, however, the discussion does not clearly differentiate how much of the observed gene expression is bacimethrin-specific versus broader Eutrophication /cyano HAB effects. Stronger evidence separating correlation from causation is needed.

The chemical quantification of bacimethrin and thiamin should include detection limits, calibration methods, and reproducibility for LC-MS/MS measurements.

The metatranscriptomics is briefly mentioned, add details on read mapping thresholds, normalization, and statistical treatment (e.g., correction for multiple testing in LEfSe analyses).

The conclusion that bacimethrin confers a competitive advantage to prototrophs (e.g., cyanoHAB taxa) is plausible, but the manuscript does not present direct experimental evidence (e.g., controlled culture assays with bacimethrin). This should be acknowledged as a limitation.

Alternative explanations such as nutrient limitation, redox dynamics, or co-occurring toxins could also structure microbial communities. The discussion should weigh these factors.

The manuscript uses a large dataset (metabolites, community composition, transcripts), but integration sometimes feels descriptive rather than mechanistic. For example, correlations between bacimethrin and gene expression are presented, but no causal or predictive modeling is attempted.

The claim that bacimethrin is “nearly equimolar” with HMP may be misleading given the high variability and large error ranges in Table 1. Statistical support for “equimolar” is weak.

Redundancy analysis (RDA) is included, but effect sizes are low, raising questions about biological significance.

The introduction is strong and well-referenced, but could more explicitly state the hypothesis: is the expectation that bacimethrin actively suppresses auxotrophs, or that it is merely a biomarker of bloom dominance?

The discussion occasionally overstates certainty (e.g., “bacimethrin provides a competitive advantage”) without caveats.

Some pathway schematics (e.g., Fig. 6) are visually complex and may be simplified for clarity.

Supporting information tables (S1–S4) are extensive but not fully summarized in the main text.

“TRCs” (thiamin and related compounds) is introduced but should be used more consistently throughout.

The term “allelopathy” may imply direct inhibition; clarify that the evidence here is correlative.

Only 16 of 24 samples yielded usable RNA. This introduces bias, but the limitation is not fully acknowledged.

Metatranscriptome and 16S community analyses show inconsistent patterns (DNA shows environment/season as stronger drivers; RNA shows cyanoHAB-driven changes). This contradiction is under-discussed.

The presence of biosynthesis/salvage genes in reference genomes is used to infer ecological function, but no expression or enzyme activity data confirm functionality.

Some tests (e.g., Wilcoxon signed-rank, PERMANOVA) are mentioned, but exact test statistics, effect sizes, and corrections for multiple testing are not always reported.

Revise to emphasize limitations and uncertainties (lack of experimental causation, possible confounders).

Strengthen methodological transparency for chemical and transcriptomic analyses.

Clarify the central hypothesis and refine the narrative to avoid overstated conclusions.

Simplify and better integrate figures/tables with the text.

Some citations like HMP auxotrophy studies are outdated, try to add recent ones.

6. PLOS authors have the option to publish the peer review history of their article (what does this mean? ). If published, this will include your full peer review and any attached files.

**Do you want your identity to be public for this peer review?** For information about this choice, including consent withdrawal, please see our Privacy Policy .

Reviewer #1: No

Reviewer #2: No

---

## [Author Response · Author response to Decision Letter 1]

30 Sep 2025

Response to Reviewers

We thank the reviewers for taking the time to comprehensively review our manuscript. We have responded to each comment below and have made necessary changes to the manuscript text and figures where appropriate and have also made minor grammatical changes in the main text where necessary. Overall, we went through and toned down our claims about bacimethrin to appropriately reflect our results being correlative rather than causative in nature. We added details throughout the manuscript where tables and figures were cited to make it easier for the reader to discern which parts of the figure/table was being referenced by the text and to better integrate the figures/tables with the text. As recommended, we did our best to clearly state any biases in our data and results and alternative explanations for our findings. We also simplified pathway diagrams for better interpretability and added all recommended statistical details when reporting results. Finally, we have added the direct web link to the Cosmos-Hub documented methods, which describe in detail all steps of the metatranscriptomics analysis performed by their software to increase the transparency of these methods. We hope that with these changes our manuscript now fully reflects the nature of our data.

Reviewer #1:

In the manuscript “The allelopathic vitamin B1 antagonist bacimethrin impacts microbial gene expression in a hypereutrophic watershed dominated by cyanobacterial blooms” it is clearly indicated that increased cyanobacterial bloom abundance is associated with increased concentrations of the thiamine and its antagonist, bacimethrin, and increased expression of thiamine biosynthesis genes (thiC, thiF, and dxs). However, some minor errors and obscurities need to be clarified before publishing.

1. The year "2025" in the Figure 1B, C caption seems to be a typographical error. Because the Materials and Methods section states that “samples were collected May 1-3 and August 28-30, 2023, during daylight”.

Yes, this was indeed a typographical error, and we have changed the year in the caption to 2023, thank you for catching this. These data were compiled from USGS publicly available data from the year of 2023.

2. Although linear regressions show that cyanoHAB abundance significantly predicts HMP and bacimethrin concentrations. But the low adjusted R² values (0.18 for HMP and 0.38 for bacimethrin) indicate that there are also other factors controlling the levels of these two chemicals. And authors were unable to identify potential genes responsible for bacimethrin synthesis. However, previous study in M. aeruginosa has suggested that genes encoding glycosyltransferase, methyltransferase, and thiaminase I may be involved in bacimethrin production, like their homologs in the bcm gene cluster found in Clostridium botulinum.

It’s completely expected that these adjusted R² values would be low because of the extreme number of microorganisms that exchange these compounds at any given time and, as pointed out, we were unable to determine the exact genes responsible for bacimethrin (though we showed data on HMP production genes). One could imagine needing to account for every microbial population that exchanges these compounds (plus every abiotic mechanism that degrades them, known and unknown) to truly be able to account for the total explained variance. The abundance of cyanoHABs are shown to be indicative of an overall microbial community state that influences their concentrations, however, there are potentially millions of individual cells exchanging the compounds, leading to their net concentrations. For this reason, a broad cellular abundance measurement will always only explain so much of a TRC concentration. We have added some of this description to the text, which now says: “Linear regression results also showed that cyanoHAB abundance significantly predicted levels of bacimethrin and its non-toxic analog HMP (Figure 2B,C). This result provided evidence that microbial community states characterized by a dominance of cyanoHABs were positively correlated with concentrations of HMP and bacimethrin. The relatively low adjusted R2 values associated with these linear regression results (Figure 2B,C) could be explained by the extreme number of individual microbes that exchange these compounds at any given time, which challenges the ability for any one factor to explain the majority in the variance of their concentrations”. As for the bacimethrin biosynthesis genes, as stated in the manuscript, we did not detect thiaminase I genes and all other genes had too many overlapping functions and were thus too nonspecific to be precise targets.

3. Why did you set a threshold of >1,000 cells mL⁻¹ for CyanoHAB high and low?

Typically, cyanoHAB thresholds are set based on the concentrations of toxins or cell abundances relevant to their threat to humans, animals, and recreation. However, these thresholds are not necessarily relevant to microbial ecology. To bin these samples, we therefore had to choose a numerical threshold that was ecologically relevant. The 75th percentile of cyanoHAB abundances was 1,453 cells/mL and all six August samples currently binned into the “High” abundance category contained cyanoHAB abundances >1,453 cells/mL. So, 1,000 was somewhat arbitrary since no good threshold exists, but meets the standard of binning these samples by the 75th percentile. We have put the 75th percentile values into the text.

Reviewer #2:

The study establishes bacimethrin as an allelopathic factor influencing microbial gene expression, however, the discussion does not clearly differentiate how much of the observed gene expression is bacimethrin-specific versus broader Eutrophication /cyano HAB effects. Stronger evidence separating correlation from causation is needed.

1. The chemical quantification of bacimethrin and thiamin should include detection limits, calibration methods, and reproducibility for LC-MS/MS measurements.

We have edited the methods section to include the following information:

The limit of detection for bacimethrin is 2.21nM, which was calculated as three times the standard deviation of the lowest standard used in analysis. Environmental values are reported in the pM range while the LOD is in the nM range due to the approximately 6 order of magnitude concentration factor produced by the SPE procedure (Suffridge 2020).

An internal standard (13C-labeled thiamin) and external standard curves were used for calibration and quantification

The TRC values presented in this manuscript are the means of three technical replicates. We have added the standard deviation of these replicates to Table S1.

2. The metatranscriptomics is briefly mentioned, add details on read mapping thresholds, normalization, and statistical treatment (e.g., correction for multiple testing in LEfSe analyses).

Some of this information is already detailed in the Supp. Methods, however, Cosmos-Hub has publicly available documentation containing all detailed information on their omics pipeline (https://docs.cosmosid.com/docs/about). This link has been added to the main text Methods as to avoid plagiarizing company documentation and to still provide all of these details. We have also added details to the main text Methods about the LEfSe analysis performed with Cosmos-Hub – based on Kruskal-Wallis sum-rank tests to detect features of significant abundance – and have mentioned that these details are similarly available at the same link as above. We did not perform corrections for multiple testing with LEfSe (now noted in the Methods) because significant results from this analysis showed clear backing by relative abundance results. For example, Limnohabitans spp. that were enriched with samples of low cyanoHAB abundance were nearly absent from samples of high cyanoHAB abundance, which we then connected to reference genome contents of these significant taxa. Another example is that picocyanobacteria that were enriched in high cyanoHAB samples with LEfSe were also high in relative abundance, based on 16S and metatranscriptomes, in these same samples.

3. The conclusion that bacimethrin confers a competitive advantage to prototrophs (e.g., cyanoHAB taxa) is plausible, but the manuscript does not present direct experimental evidence (e.g., controlled culture assays with bacimethrin). This should be acknowledged as a limitation.

We agree and have noted this limitation in the Discussion.

4. Alternative explanations such as nutrient limitation, redox dynamics, or co-occurring toxins could also structure microbial communities. The discussion should weigh these factors.

We have added description to the Discussion about how nutrients and redox dynamics could influence microbial communities. However, as mentioned in the manuscript, we did not find evidence for the expression of cyanobacterial toxin genes, so we did not include this as a possible factor influencing microbial communities.

5. The manuscript uses a large dataset (metabolites, community composition, transcripts), but integration sometimes feels descriptive rather than mechanistic. For example, correlations between bacimethrin and gene expression are presented, but no causal or predictive modeling is attempted.

We agree that our study does not attempt to model these conditions to predict conditions in different watersheds or time periods. This is inherent to our data because this is the first time freshwater microbial ecology has been linked to bacimethrin allelopathy and connecting so many disparate datapoints is therefore correlative in nature, given the lack of past data on this subject. Our goal is for our results to be a jumping off point for future mechanistic and causative research on how bacimethrin allelopathy in natural ecosystems influences microbial ecology.

6. The claim that bacimethrin is “nearly equimolar” with HMP may be misleading given the high variability and large error ranges in Table 1. Statistical support for “equimolar” is weak.

We have removed this claim and instead mentioned that the concentrations of these compounds are similar.

7. Redundancy analysis (RDA) is included, but effect sizes are low, raising questions about biological significance.

It’s unsurprising, given the extreme number of factors that, in reality, influence microbial community compositions (interactions between thousands of microbial populations, the sheer complexity of metabolomes exchanged by those populations, auxotrophies beyond those examined here for other vitamins/vitamin precursors and amino acids, the availability of trace metals, etc.), that the effect size is small. It is standard in microbial ecology to relate few environmental/chemical factors with microbial communities, knowing that multitudes of other factors are simultaneously at play. For this reason, we do not think this shows a lack of support for biological significance, but rather that TRCs are some of many chemical factors that influence complex microbial communities. Plus, RDA prevents the use of a large number of environmental factors because it leads to variable dependence and will increase variance inflation factors, which we checked (see SI).

8. The introduction is strong and well-referenced, but could more explicitly state the hypothesis: is the expectation that bacimethrin actively suppresses auxotrophs, or that it is merely a biomarker of bloom dominance?

We agree and have added a hypothesis to at the end of the Introduction prior to our listed objectives.

9. The discussion occasionally overstates certainty (e.g., “bacimethrin provides a competitive advantage”) without caveats.

We agree and have toned down this claim and emphasized the fact that cyanoHABs use alternate mechanisms to outcompete other bacterioplankton. We also adjusted the title wording to reflect the existence of uncertainty in our initial claim that alluded to bacimethrin directly influencing gene expression and thus providing competitive advantage.

10. Some pathway schematics (e.g., Fig. 6) are visually complex and may be simplified for clarity.

We have simplified these Fig. 6 pathways by eliminating the two panels, reducing text, and showing one single generic photoautotroph interacting (with split biochemical scenarios depending on cyanoHAB impact) with two different generic heterotrophs under the two cyanoHAB impact scenarios. We simplified Figure 3A by omitting the inclusion of TRC phosphorylation steps in the pathway, and directed the reader to Jurgenson et al. 2009 in the figure caption for a more detailed pathway. We also clarified that the genes shown were those that were annotated in our dataset.

11. Supporting information tables (S1–S4) are extensive but not fully summarized in the main text.

We have added a section in the SI that describe these tables with more detail along with table captions. We have also added more description when these tables are cited in the main text to relevant table components where necessary.

12. “TRCs” (thiamin and related compounds) is introduced but should be used more consistently throughout.

We agree and have added the acronym more consistently throughout the text, but use individual names of TRCs where necessary.

13. The term “allelopathy” may imply direct inhibition; clarify that the evidence here is correlative.

We have made this clarification at the beginning of the Discussion.

14. Only 16 of 24 samples yielded usable RNA. This introduces bias, but the limitation is not fully acknowledged.

Of course, this technically biases results towards samples with high enough microbial gene expression to be picked up by NextSeq, but this is the limit of our current technology (Illumina prep. kits only work with a certain, strict threshold of input RNA). We actually troubleshooted with Illumina and the Oregon State Center for Quantitative Life Sciences just to get the 16 to work; mRNA is extremely challenging to extract and sequence from water samples. We have noted this bias in the Methods and stated that we were still able to capture samples across the main ecosystem types of interest other than spring water. This result also wasn’t surprising because spring water and surface waters that were unimpacted by cyanoHABs were likely to contain too low of microbial biomasses (see Figure S2) to yield enough mRNA from transcriptionally active cells to make it through the main steps of the workflow: RNA extraction, cleaning and concentrating, Illumina prep., and NextSeq.

15. Metatranscriptome and 16S community analyses show inconsistent patterns (DNA shows environment/season as stronger drivers; RNA shows cyanoHAB-driven changes). This contradiction is under-discussed.

We agree and have added a description of this into the Results. This gets at the nature of 16S v. mRNA-seq. DNA sticks around in the ecosystem and can be measured in unviable/inactive cells whereas mRNA is ephemeral and active gene expression must be occurring for it to be extracted in a high enough concentration to be measured as sequencing results, and the influence of rare taxa v. abundant taxa. We had originally removed discussion of this result for brevity.

16. The presence of biosynthesis/salvage genes in reference genomes is used to infer ecological function, but no expression or enzyme activity data confirm functionality.

We agree, but we did not have the ability to look at enzyme activity data and, as noted in the text, certain salvage genes could only be queried in reference genomes because many of them are custom HMMs from previous published work not available in Cosmos-Hub functional reference databases. We have now added as a statement in the Discussion that future microbial ecophysiology research, similar to Yazdani et al, (2025), is needed to confirm our correlative findings.

17. Some tests (e.g., Wilcoxon signed-rank, PERMANOVA) are mentioned, but exact test statistics, effect sizes, and corrections for multiple testing are not always reported.

We have gone through and reported all details for these tests. Wilcoxon signed-rank and PERMANOVA tests did not have corrections for multiple testing (now noted in the Methods and Results). No Wilcoxon signed-rank tests were significant and it’s now repor

---

## [Decision Letter · Decision Letter 1]

16 Oct 2025

Bacimethrin, an allelopathic vitamin B1 antagonist, is linked with microbial gene expression patterns in a hypereutrophic watershed

PONE-D-25-38532R1

Dear Dr. Christopher,

We’re pleased to inform you that your manuscript has been judged scientifically suitable for publication and will be formally accepted for publication once it meets all outstanding technical requirements.

Kind regards,

Barathan Balaji Prasath

Academic Editor

PLOS ONE

Additional Editor Comments (optional):

Reviewers' comments:

Reviewer's Responses to Questions

**Comments to the Author**

1. If the authors have adequately addressed your comments raised in a previous round of review and you feel that this manuscript is now acceptable for publication, you may indicate that here to bypass the “Comments to the Author” section, enter your conflict of interest statement in the “Confidential to Editor” section, and submit your "Accept" recommendation.

Reviewer #1: All comments have been addressed

Reviewer #2: All comments have been addressed

2. Is the manuscript technically sound, and do the data support the conclusions?

Reviewer #1: Yes

Reviewer #2: Yes

3. Has the statistical analysis been performed appropriately and rigorously? 

Reviewer #1: Yes

Reviewer #2: Yes

4. Have the authors made all data underlying the findings in their manuscript fully available?

Reviewer #1: Yes

Reviewer #2: Yes

5. Is the manuscript presented in an intelligible fashion and written in standard English?

Reviewer #1: Yes

Reviewer #2: Yes

6. Review Comments to the Author

Reviewer #1: The manuscript has been properly revised and all comments have been addressed, and the manuscript is ready for acceptance and I would like to thank the respected editor for trusting me and assigning me the review of this manuscript.

Reviewer #2: Authors have successfully addressed my previous comments, resulting in a thoroughly improved and well-revised manuscript that demonstrates significant enhancement in clarity and quality.

7. PLOS authors have the option to publish the peer review history of their article (what does this mean? ). If published, this will include your full peer review and any attached files.

**Do you want your identity to be public for this peer review?** For information about this choice, including consent withdrawal, please see our Privacy Policy .

Reviewer #1: No

Reviewer #2: No

---

## [Editor Report · Acceptance letter]

PONE-D-25-38532R1

PLOS ONE

Dear Dr. Suffridge,

I'm pleased to inform you that your manuscript has been deemed suitable for publication in PLOS ONE. Congratulations! Your manuscript is now being handed over to our production team.

Kind regards,

on behalf of

Dr. Barathan Balaji Prasath

Academic Editor

PLOS ONE